# Structural basis for ALK2/BMPR2 receptor complex signaling through kinase domain oligomerization

Christopher Agnew[1,8], Pelin Ayaz[2,8], Risa Kashima [1], Hanna S. Loving[3], Prajakta Ghatpande[1], Jennifer E. Kung [1,7], Eric S. Underbakke [3✉], Yibing Shan [2✉], David E. Shaw [2,4✉], Akiko Hata [1,5] & Natalia Jura [1,6✉]

Upon ligand binding, bone morphogenetic protein (BMP) receptors form active tetrameric complexes, comprised of two type I and two type II receptors, which then transmit signals to SMAD proteins. The link between receptor tetramerization and the mechanism of kinase activation, however, has not been elucidated. Here, using hydrogen deuterium exchange mass spectrometry (HDX-MS), small angle X-ray scattering (SAXS) and molecular dynamics (MD) simulations, combined with analysis of SMAD signaling, we show that the kinase domain of the type I receptor ALK2 and type II receptor BMPR2 form a heterodimeric complex via their C-terminal lobes. Formation of this dimer is essential for ligand-induced receptor signaling and is targeted by mutations in BMPR2 in patients with pulmonary arterial hypertension (PAH). We further show that the type I/type II kinase domain heterodimer serves as the scaffold for assembly of the active tetrameric receptor complexes to enable phosphorylation of the GS domain and activation of SMADs.

[1] Cardiovascular Research Institute, University of California San Francisco, San Francisco, CA, USA. [2] D. E. Shaw Research, New York, NY, USA. [3] Roy J. Carver Department of Biochemistry, Biophysics, and Molecular Biology, Iowa State University, Ames, IA, USA. [4] Department of Biochemistry and Molecular Biophysics, Columbia University, New York, NY, USA. [5] Department of Biochemistry and Biophysics, University of California San Francisco, San Francisco, CA, USA. [6] Department of Cellular and Molecular Pharmacology, University of California San Francisco, San Francisco, CA, USA. [7] Present address: Department of Structural Biology, Genentech, Inc., South San Francisco, USA. [8] These authors contributed equally: Christopher Agnew, Pelin Ayaz. ✉email: esu@iastate.edu; Yibing.Shan@DEShawResearch.com; David.Shaw@DEShawResearch.com; Natalia.Jura@ucsf.edu

Signaling by transmembrane receptor kinases, which are composed of an extracellular ligand-binding domain, a single transmembrane helix, and an intracellular kinase domain, is fundamental to organismal development and adult homeostasis. Two different classes of such receptors exist: receptor tyrosine kinases (RTKs) and receptor serine/threonine kinases (RSTKs). While the RTKs encompass several unrelated sub-families of receptors, all RSTKs belong to the transforming growth factor-β (TGFβ) superfamily of receptors, which include the TGFβ receptors and the bone morphogenetic protein (BMP) receptors. TGFβ and BMP receptors share a significant degree of homology in their extracellular and intracellular domains and are subdivided into two distinct groups of receptors: type I and type II receptors. Signaling is activated when two types I and two type II receptors are brought into proximity to form a tetramer by dimeric ligands that associate with the extracellular domains of the type I and type II receptors. Formation of the type I/type II tetramer promotes transphosphorylation of the intracellular portions of the receptors, resulting in recruitment and activation of the receptor-specific SMAD (R-SMAD) transcription factors. This tetrameric receptor arrangement around a dimeric ligand is unique to the RSTKs and significantly differs from known mechanisms of RTK activation, which tend to activate via the formation of dimeric complexes upon growth factor binding.

The BMP receptors are important for a diverse range of cellular functions, including embryonic development, ossification, neurogenesis, tissue patterning, and homeostasis[1]. There are four type I and three type II BMP receptors, which differ slightly in their tissue distribution, preference for ligands, and oligomerization partners[2]. The type I BMP receptors, commonly known as activin-like kinases (ALKs), consist of ALK1 (also known as ACVRL1), ALK2 (also known as ACVR1 and ActRIA), ALK3 (also known as BMPRIA), and ALK6 (also known as BMPRIB). The type II BMP receptors, consist of BMP receptor type II (BMPR2) and activin type II receptors A and B (ActRIIa/ACVR2a and ActRIIb/ACVR2b). Based on sequence homology, the type II BMP receptors can be further divided into two subclasses. The first contains BMPR2, which binds BMP and growth differentiating factor (GDF) ligands. The second subclass encompasses more promiscuous ACVR2a and ACVR2b, which predominantly interact with activins, but can also bind BMPs and GDFs. Specific ligand and receptor combinations determine the signaling output of BMP receptor activation, with a high level of complexity encoded in the different possible pairing arrangements[3,4]. Further complexity is added by the BMP ligands, which can also heterodimerize leading to formation of less canonical receptor tetramers[5–7], and by a variety of transmembrane co-receptors—collectively called type III receptors, which further diversify the signaling spectrum of the BMP receptors[8].

The canonical view of BMP receptor activation is that of a complex formed between type-I receptor homodimer and type-II receptor homodimer stabilized by the dimeric ligand[9,10]. However, crystallographic studies on the extracellular portions of the receptors show that type I and type II receptors do not engage in homodimeric interactions in the active complex. The small extracellular domains of these receptors (~14 kDa) have a disulfide-bonded three-finger toxin fold and contain one ligand-binding site[8]. Each ligand has distinct type I and type II receptor binding sites, which are of equal binding affinity to the receptors in BMP ligands[11]. A dimeric ligand scaffolds four extracellular domains of the receptors into a symmetric, tetrameric structure in which the type I receptors are positioned diagonally opposite each other, which is mirrored by the arrangement of the type II receptors (Fig. 1a). The juxtamembrane domains, which connect the intracellular domains and transmembrane domains, are short in length in BMP receptors, and therefore limit the degree to which the kinase domains can reorient and interact in the active

tetramer. Thus, the alternating order of the extracellular domains of type-I and type-II receptors in the ligand-bound tetramer imposes a similar geometry on the intracellular domains, which is predicted to favor the heterodimeric kinase interactions.

Activation of the type I kinase domain is a regulated process involving multiple control systems that prevent nonspecific signaling. In the active receptor complex, the type II receptor kinase phosphorylates and activates the type I receptor. Upon its activation, the type I receptor then phosphorylates R-SMADs. In the inactive state, the type I kinase is autoinhibited by the GS domain, a short glycine and serine-rich region directly N-terminal to the kinase. The GS-domain binds to the kinase N-lobe and locks the catalytic helix C in an inactive conformation[12]. This interaction is further enhanced by the inhibitory protein FKBP12, which docks on top of the GS domain[12,13]. Additional regulation of kinase activity is provided by the intracellular inhibitory SMADs, which bind to type I kinases and inhibit activation, and through posttranslational modifications which regulate receptor trafficking, degradation, and activation[14]. All these inhibitory constraints are temporarily released during ligand-induced activation, though the mechanistic details of these processes remain poorly defined. One of the critical steps is the activation of the type II kinase, because it phosphorylates the GS domain, which is necessary for the disengagement of that domain from the type I kinase[9,15]. The phosphorylated type I kinase adopts an active conformation, enabling phosphorylation of an R-SMAD, which subsequently forms a heteromeric complex with a co-SMAD, SMAD4[4]. The SMAD complex then translocates to the nucleus, where it binds promoter regions of target genes and modulates transcription[4].

Deregulation of BMP receptors results in human diseases[16]. Gain-of-function mutations in the type I receptor ALK2 cause fibrodysplasia ossificans progressiva (FOP) and diffuse intrinsic pontine glioma (DIPG)[17]. These mutations predominantly map to the GS domain/kinase domain interface and are positioned to disrupt their autoinhibitory interaction[17,18]. A constitutively active ALK2 GS domain mutant (Q207D), which is similar to an FOP mutation (Q207E), acts independently of ligand binding but requires interaction with a type II receptor to be fully active regardless of whether the type II kinase is active or inactive. This indicates that type II receptors have an important scaffolding function within the active receptor complex[19]. Loss-of-function or -expression mutations in BMPR2 result in pulmonary arterial hypertension (PAH), a disease of the pulmonary arteries with high morbidity and mortality[20]. The inactivating BMPR2 mutations in PAH are located throughout the protein, including the extracellular domain, kinase domain, and the C-terminal tail domain[20]. Mutations in the kinase domain disrupt signaling with little to no effect on catalytic activity[21] suggesting that in addition to the catalytic activity, the BMPR2 kinase domain also carries an additional regulatory function that has not been appreciated previously.

While a tetrameric architecture of the extracellular domain complex has been reported previously based on the crystal structures of different type I/ type II receptor combinations and their ligands (summarized in ref. [8]), no structural insights into the organization of the intracellular domains in the active complex exist. It is unknown how the tetrameric kinase complex forms and what its functional significance in terms of kinase activation and downstream signaling might be. Why do four kinases have to be recruited in a complex to initiate signaling? Our knowledge of receptor organization prior to ligand stimulation points to a complex array of interactions that cannot be explained by a simple mechanism. Unliganded BMP receptors are thought to predominantly form homodimers at the cell surface, however, their existence as monomers or preformed heteromeric complexes between type I and type II receptors has also been

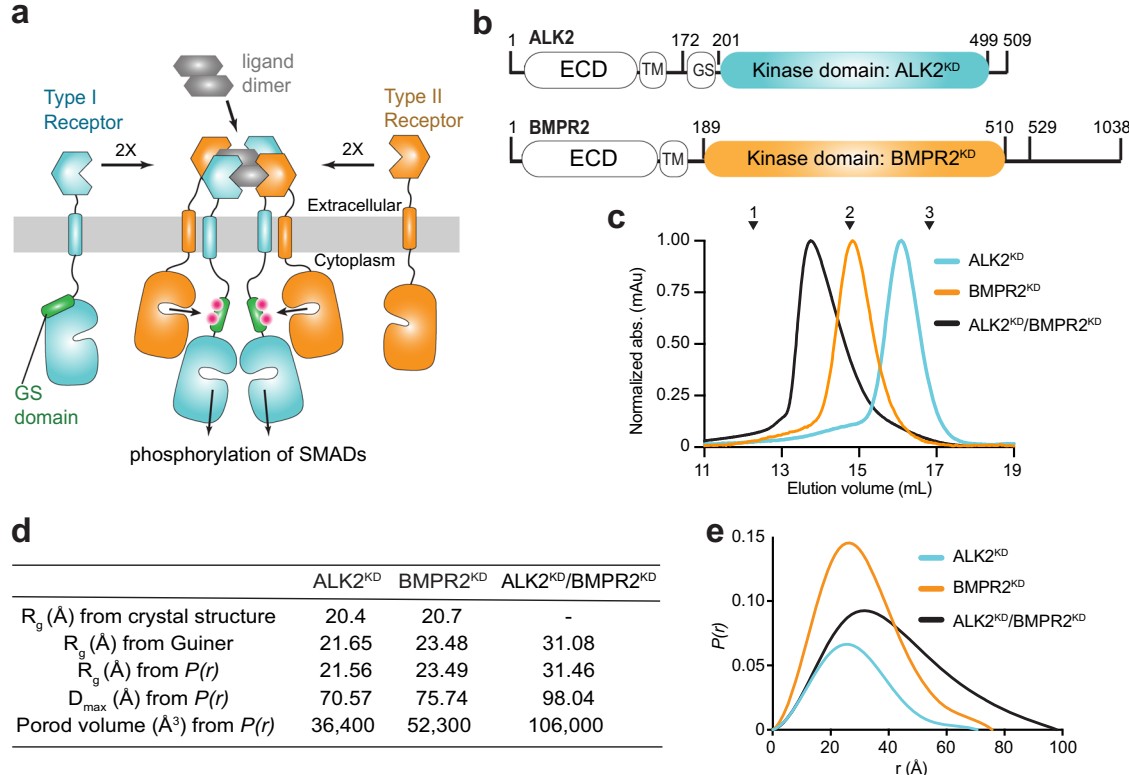

**Fig. 1 The BMPR2 and ALK2 kinase domains form a thermodynamically stable heterodimeric complex in solution. a** Schematic diagram of the ligand-induced type I/type II receptor tetramer and resulting regulatory Gly/Ser rich (GS) domain phosphorylation. Pink dots symbolize phosphorylated residues within the GS domain. **b** Diagram depicting the domain boundaries within the ALK2 and BMPR2 receptors, including the extracellular domain (ECD), transmembrane (TM) helix, the GS domain, and the kinase domains (KD). Residues marking kinase domain boundaries and boundaries of the constructs used in this study are marked. **c** Size-exclusion chromatograms of ALK2$^{KD}$ (cyan), BMPR2$^{KD}$ (orange) and the ALK2$^{KD}$/BMPR2$^{KD}$ complex (black) resolved on a Superdex 200 Increase 10/300 GL column. Molecular weight standards are indicated above: 1— γ-globulin (158,000 Da), 2—ovalbumin (44,000 Da) and 3—myoglobin (17,000 Da). **d** The radius of gyration ($R_g$) calculated from the crystal structures of ALK2 (PDB ID: 3MTF) and BMPR2 (PDB ID: 3G2F) compared with molecular dimensions calculated from the SEC-SAXS profiles of the ALK2$^{KD}$, BMPR2$^{KD}$, and the ALK2$^{KD}$/BMPR2$^{KD}$ complex. **e** SAXS P(r) function of ALK2$^{KD}$ (cyan), BMPR2$^{KD}$ (orange), and the ALK2$^{KD}$/BMPR2$^{KD}$ (black) complex.

reported[22,23]. If the type I and type II receptors form heterodimers prior to ligand binding, it is unclear how they would avoid premature activation via type II-mediated phosphorylation of the GS domain.

In this study, we describe a heterodimeric complex formed by the kinase domains of type I receptor ALK2 and type II receptor BMPR2, which engages the C-terminal lobes of both kinases. In this orientation, the N-terminal GS domain of ALK2, which inhibits the kinase when unphosphorylated, is positioned away from the BMPR2 active site. As a result, BMPR2 is actively prevented from phosphorylating the GS domain impeding erroneous activation of ALK2. Using molecular dynamics (MD) simulations, we show that two such autoinhibited ALK2/BMPR2 heterodimers can interact across a newly identified interface that engages the N-lobes of the type-I and type-II kinases, resulting in the dissociation of the GS domains of the type I kinases and their subsequent positioning into the active sites of the type II kinases for phosphorylation. Using SMAD signaling assays, we demonstrate the functional significance of the C-lobe and N-lobe-mediated interfaces for signaling. Our findings provide a mechanistic explanation for a number of PAH mutations localized in the kinase domain in BMPR2, which we show inactivate BMPR2 signaling by disrupting the C-lobe interface. This is the first oligomeric model explaining interactions between type I and type II kinases in an active tetrameric complex that provides mechanistic justification for the need of two copies of each kinase type.

## Results

**The ALK2 and BMPR2 kinase domains form a heterodimer in solution.** We expressed and purified kinase domains of the type I receptor, ALK2 (ALK2$^{KD}$, residues 201–499) and the type II receptor, BMPR2 (BMPR2$^{KD}$, residues 189–529) (Fig. 1b). Size exclusion chromatography (SEC) profiles of ALK2$^{KD}$ and BMPR2$^{KD}$ alone are consistent with their monomeric state, despite a notable difference in the elution volume, as previously reported[19]. A stoichiometric mixture of ALK2$^{KD}$/BMPR2$^{KD}$ forms a thermodynamically stable heteromeric complex (Fig. 1c). Such a complex has previously been observed when BMPR2$^{KD}$ was mixed with the ALK2 kinase domain containing the regulatory GS domain (GS-ALK2$^{KD}$)[19]. To determine the oligomeric state of the ALK2$^{KD}$/BMPR2$^{KD}$ complex we used size-exclusion chromatography—small-angle X-ray scattering (SEC-SAXS). Despite the notable difference in elution volumes for ALK2$^{KD}$ and BMPR2$^{KD}$ constructs in SEC analysis, both kinases yielded particles that are equivalent in size and shape to kinase monomers by SEC-SAXS (Fig. 1d, e). A mixture of ALK2$^{KD}$ and BMPR2$^{KD}$ was consistent with particles of the approximate size for a compact kinase dimer (Fig. 1d, e and Supplementary Fig. 1).

**Generation of ALK2 and BMPR2 kinase domain dimer models.** The heterodimeric ALK2$^{KD}$/BMPR2$^{KD}$ complex was not amenable to crystallization. Thus, in order to generate a structural model that could be tested experimentally, we conducted

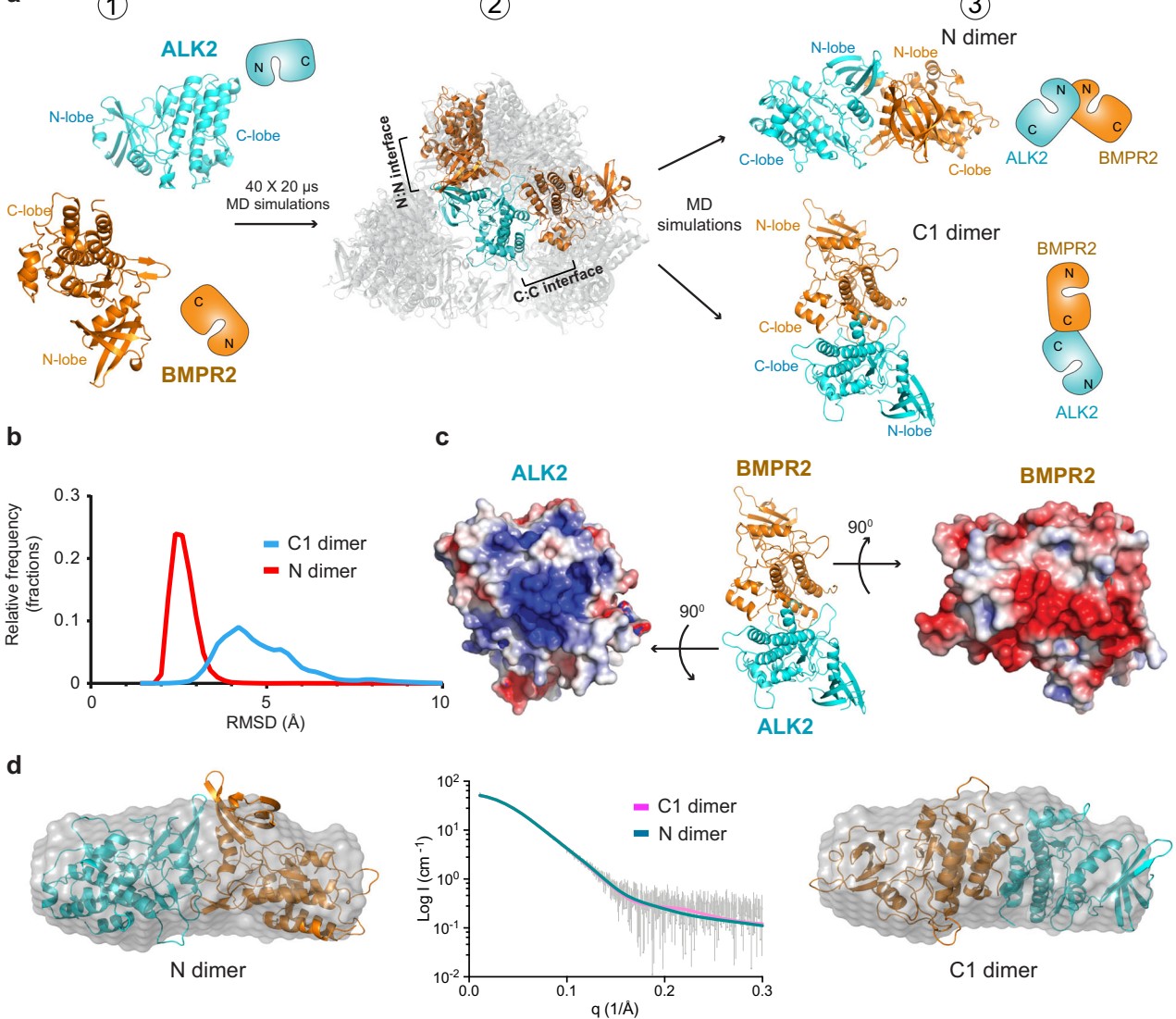

**Fig. 2 Generation of the ALK2/BMPR2 kinase heterodimer model. a** Generation of the models: (1) Starting positions of ALK2 kinase domain (cyan (PDB 3H9R); residues 202–499) and BMPR2 kinase domain (orange (PDB 3G2F); residues 197–510). Their minimum separation is 15 Å. (2) The poses that were conformationally stable for more than 5 μs in the 40 independent 20 μs binding simulations superposed on ALK2 kinase domain. The two poses that were used in subsequent modeling steps are shown in solid colors. BMPR2 is distributed almost spherically around ALK2, but it is rendered as partially transparent so that ALK2 is visible. (3) Final two poses after further simulations of the N dimer and the C1 dimer (Supplementary Table 5). **b** Normalized distribution of the RMSDs with respect to the initial conformations in simulations of the C1 dimer (blue) and N dimer (red) models. The RMSDs were calculated using all Cα atoms of the two kinase domains after superposition to the initial conformation using these atoms. The raw RMSD data from the simulations are plotted in Supplementary Fig. 11a, c. **c** Electrostatic surface potential calculated using APBS[94] at the C1-dimer interface at the C-lobes of the ALK2 and BMPR2 kinases. **d**, Scattering data for the ALK2$^{KD}$/BMPR2$^{KD}$ complex plotted with CRYSOL fits of the N dimer (teal line; $\chi^2 = 0.212$) and the C1 dimer (pink line; $\chi^2 = 0.215$). Ab initio bead model for the ALK2$^{KD}$/BMPR2$^{KD}$ complex is displayed with either the N or C1 dimer superimposed.

unbiased MD simulations in a way analogous to a previous study that led to the elucidation of JAK2 kinase/pseudokinase domain interactions[24,25]. The atomic structures of the ALK2 kinase (without the GS domain) and the BMPR2 kinase were placed in a non-contacting pose inside a box of explicit solvent molecules (Fig. 2a, state 1). Forty independent simulations were run for 20 μs each, each resulting in a final pose in which the two kinase domains were in contact (Fig. 2a, state 2). Inspection of the structural poses that remained unchanged for at least 5 μs in the simulations showed a variety of possible interactions for the ALK2-BMPR2 kinase dimer. Out of a total of 18 such poses, 8 feature small buried surface area (<800 Å²) at the interfaces (see Supplementary Table 1) and 4 feature interfaces that primarily

involve flexible loop regions. These 12 poses were deemed unlikely to be thermodynamically stable and were discarded. Of the six remaining poses, five feature interfaces involving the two N-lobes, with their C helices interacting with one another, and one features an interface involving the C-lobes.

Of these six poses, two had interesting characteristics that prompted further examination. In the first pose, ALK2 and BMPR2 kinases interacted across the N-terminal lobes (N-lobes), forming a dimer we refer to as an *N-lobe/N-lobe dimer*, or more briefly, the *N dimer* (Fig. 2a, state 3). The N dimer was interesting, compared with the other 4 complexes with interfaces involving the N-lobes and the αC helices, for several reasons: (i) it was symmetric in geometry (Fig. 2b); (ii) the pose was

conformationally remarkably stable in simulation (Supplementary Fig. 2a, lower right panel); (iii) the dimer interface on the ALK2 kinase domain overlapped with the binding site for the GS-domain and FKBP12, providing a potential structural explanation for activation (Supplementary Fig. 3a); (iv) the dimer interface engaged helix C on both kinases, representing a possible mode for regulation of kinase activity (Supplementary Fig. 3b); and (v) the N-terminus of the ALK2 kinase domain in this dimer was positioned in proximity to the BMPR2 active site providing a possible path for the presentation of the ALK2 GS domain for phosphorylation by the BMPR2 kinase (Supplementary Fig. 3c).

In the second pose we focused on, ALK2 and BMPR2 kinases interacted by way of their C-terminal lobes (C-lobes), engaging primarily helices H and I of the BMPR2 kinase and helices G and H of the ALK2 kinase. This ALK2-BMPR2 dimer was notable in that (i) once the kinases adopted this pose, it was largely unchanged for the remainder of the simulation (Supplementary Fig. 2a, upper right panel); and (ii) the two dimer interfaces had extensive complementary electrostatic surface potentials: negatively charged on BMPR2 and positively charged on ALK2 (Fig. 2c). We refer to this dimer as the *C-lobe/C-lobe 1 dimer*, or more briefly, the *C1 dimer* (see Supplementary Data 1 for the structural coordinates), to distinguish it from a slightly different form of C-lobe/C-lobe ALK2-BMPR2 dimer that is discussed later in this report. We note that in the 40 simulations, we also observed several ALK2/BMPR2 dimer conformations with similar C-lobe interaction surfaces but with different orientations. All of these other C-lobe-mediated interactions exhibited more conformational fluctuation than the C1 dimer (Supplementary Fig. 2). The higher conformational fluctuation of these alternate conformations is consistent with the greater degree of conformational flexibility and lower energetic penalty typically associated with protein–protein interactions dominated by electrostatic complementarity compared with interactions dominated by hydrophobic interactions and shape complementarity.

To determine if the models generated by our MD simulations were consistent with the complex identified in the solution, we compared the SAXS data from the solution-state complex with the simulation-generated structures (Fig. 2d). The N dimer and the C1 dimer fit to the scattering data (Fig. 2d, middle panel) with comparable $\chi^2$ values (0.212 and 0.215, respectively). We then generated an ab initio bead model based on the scattering data, and superimposed the N dimer and the C1 dimer structures. Both dimers fit the bead model, indicating that the solution-state complex is a kinase dimer (Fig. 2d and Supplementary Fig. 1), but the resolution of the SAXS data was too low to permit differentiation between the N and C1 dimer models.

**ALK2 and BMPR2 heterodimerize in solution via the C-lobe/C-lobe interface**. We used hydrogen-deuterium exchange mass-spectrometry (HDX-MS) to investigate how accurately the modeled poses represent the heterodimeric complex formed by ALK2 and BMPR2 in solution. H/D exchange timecourses were collected for ALK2$^{KD}$ alone, BMPR2$^{KD}$ alone, and an equimolar solution of both kinases. Deuterium incorporation of each uniquely identifiable peptic fragment was assessed using LC-MS. Suppressions of exchange rate report on local decrease of structural fluctuation or lower solvent accessibility. We observed multiple overlapping peptides in localized regions of each kinase exhibiting significantly decreased rates of deuterium exchange in the complex relative to each kinase domain alone (Supplementary Fig. 4). Strikingly, in both kinases, the protection localized to the C-lobe regions: in BMPR2$^{KD}$ at the bottom of the C-lobe, and in ALK2$^{KD}$ slightly toward the back side of the C-lobe (Fig. 3a), centrally engaging helix I (Fig. 3b). These regions largely overlap with the C1 dimer interfaces (Figs. 2a, 3b).

In the ALK2$^{KD}$, the strongest protection mapped to the final 21 residues of the C-lobe, which form helix I (residues 488–499), and an αH-αI linker that connects helix I with helix H (residues 479–487) (Supplementary Fig. 4). Peptides that fall within these regions exhibited striking decreases in exchange (i.e., >25% lower deuterium incorporation at time points sampling the middle range of the exchange regime) within helix I and in the αH-αI linker. The amphipathic helix I is almost completely composed of basic residues on one side, such as K492, K493, and R490 (Fig. 3b), and together with the αH-αI linker lies at the center of the large basic patch within the ALK2 kinase C-lobe. The basic patch surface envelops the N-terminal half of helix F (residues 396–408), which also exhibited strong protection from exchange upon binding BMPR2$^{KD}$ (Fig. 3b). This concerted decrease in exchange is indicative of the reduced structural fluctuation of this region in ALK2$^{KD}$ upon binding of the BMPR2$^{KD}$ and is consistent with the formation of the C1 ALK2/BMPR2 kinase heterodimer.

The structure of the C-lobe of BMPR2$^{KD}$ also undergoes stabilization upon dimerization with the ALK2$^{KD}$ (Fig. 3a, b, Supplementary Fig. 4). One of the protected regions encompasses the αH-αI linker (residues 485–492), a surface displaying and bordering a number of acidic residues, including E481, D482, D485, D487, and E489. This region is in direct contact with helix I on ALK2 in the C1 dimer model (Fig. 3b). In the crystal structure of the BMPR2 kinase domain, the αH-αI linker forms a partial helix and is mostly surface accessible, except for the highly conserved R491 that packs into the C-lobe core where it forms an electrostatic interaction with the sidechain of E386 and with the backbone carbonyl oxygens of Q403 and Q486 (Fig. 3c). E386 resides within a short helix, directly N-terminal to helix F, called helix EF, which exhibits a significant decrease in exchange rate upon BMPR2$^{KD}$ dimerization with ALK2$^{KD}$ (Fig. 3b). Restriction in the mobility of this region likely occurs allosterically, via stabilization of the dimer interface. The most extensive region of protection in the C-lobe of BMPR2$^{KD}$ centers on helix G and its adjacent N- and C-terminal loop regions that collectively encompass residues 418–465 (Fig. 3d). The C-terminal end of helix G and a loop connecting it to helix H (the αG-αH linker) pack directly into the interface of the C1 dimer. The engagement of the αG-αH linker of the BMPR2 kinase at the interface also likely contributes to the stabilization of the adjacent αF-αG linker (Fig. 3d). Interestingly, no detectable perturbation of exchange rates was observed at helix H of BMPR2 kinase despite its localization at the putative dimer interface (Fig. 3b). This is attributable to the high intrinsic stability of the helix, as peptides covering the H helix strongly resisted exchange in both BMPR2$^{KD}$ and the ALK2$^{KD}$/BMPR2$^{KD}$ complex. Collectively, MD and HDX-MS analyses provide an integrative model for dimerization of the ALK2 and BMPR2 kinases in solution via the C1 interface.

**The C-lobe/C-lobe ALK2/BMPR2 dimer is compatible with the GS domain binding**. Only one region of BMPR2$^{KD}$, which maps outside of the C-lobe-centered interface, is stabilized upon BMPR2$^{KD}$/ALK2$^{KD}$ heterodimerization, albeit weakly. It spans the β4-β5 strands (residues 262–276) in the N-lobe and falls within the interface of our structural N dimer model (Figs. 2a and 3a). However, no significant HDX-MS differences were measured on the ALK2$^{KD}$ side of the N dimer interface. In isolation, the lack of exchange rate perturbations does not conclusively rule out a binding interface. However, the strong C-lobe exchange perturbations argue in favor of the formation of the C-lobe dimer in

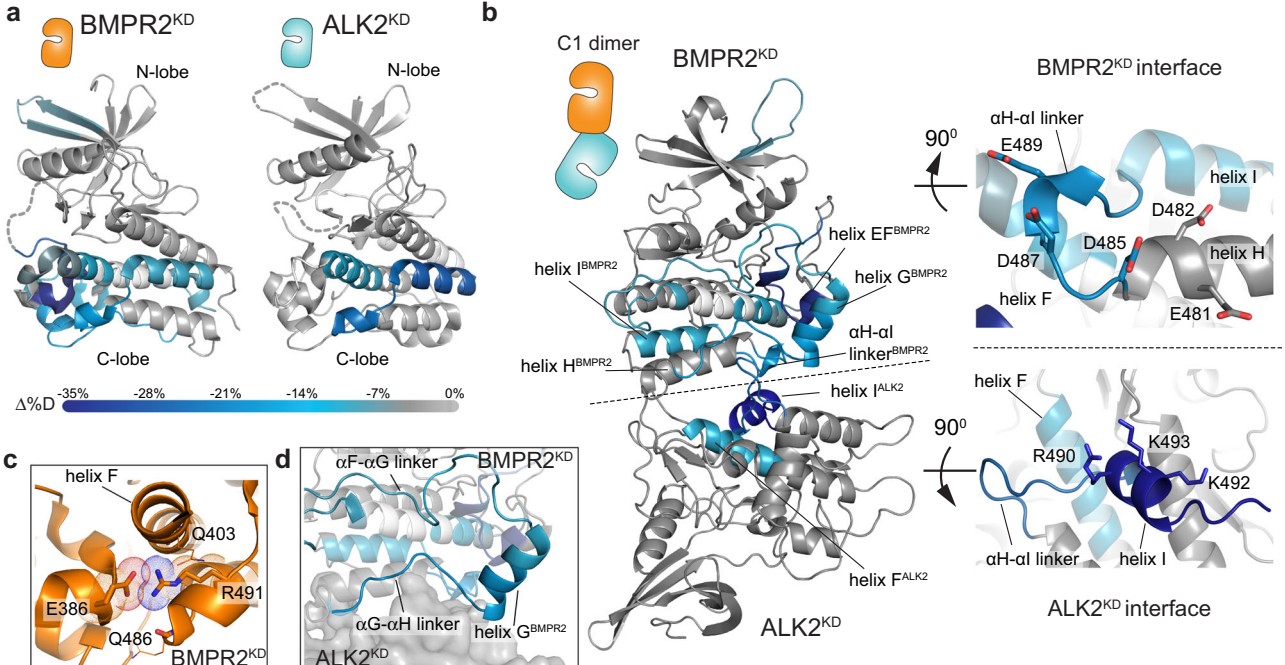

**Fig. 3 The ALK2^KD/BMPR2^KD heterodimer interface localizes to the kinase C-lobes. a** Condensed HDX-MS results comparing isolated BMPR2^KD or ALK2^KD vs. the BMPR2^KD/ALK2^KD complex are mapped to structural models of ALK2^KD (PDB ID: 3H9R) and BMPR2^KD (PDB ID: 3G2F). Complex-induced deuterium exchange perturbations are color-coded according to the scale bar (below) representing the average difference in %D incorporation (Δ%D). Statistical significance was assessed with a two-tailed, unpaired Student's *t* test. Regions exhibiting no significant differences in exchange are colored in gray. Regions lacking peptide coverage are noted in white. Complex-induced exchange perturbations mapped to primary sequence are reported in Supplementary Fig. 4. **b** Deuterium exchange perturbations induced by ALK2^KD/BMPR2^KD complex formation are mapped to the ALK2/BMPR2 C1 kinase dimer model generated by MD simulations. The insets (right) highlight putative interface surfaces. The color-coding scale is the same as in (**a**). **c** Zoomed view of the electrostatic interaction network between BMPR2 residue R491, the sidechain of E386, and the backbone carbonyl oxygens of Q403 and Q486. **d** Detailed view of the complex-induced exchange perturbations centered around helix G of the BMPR2 kinase in the context of the C1 kinase dimer model. Perturbations are color-coded as in (**a**).

solution. Corroborating the negligible role of the N dimer interface, the presence of the GS domain on ALK2 does not interfere with BMPR2^KD/ALK2^KD heterodimerization in solution (Supplementary Fig. 5)[19]. This would be expected in the N dimer since GS domain binding to the ALK2 kinase directly blocks the N dimer interface (Supplementary Fig. 3a).

To investigate whether the presence of the GS domain changes the dynamics of BMPR2 and ALK2 dimerization via the C-lobe-centered interface, we repeated the HDX-MS experiments with the ALK2 construct containing the GS domain (GS-ALK2^KD). By comparing the H/D exchange dynamics of ALK2^KD alone versus GS-ALK2^KD alone, we saw significant stabilization of the ALK2 kinase N-lobe in the presence of the GS domain, consistent with the crystallographically mapped interface[26] (Supplementary Fig. 6). Then, we compared the dynamics of GS-ALK2^KD/ BMPR2^KD complex vs ALK2^KD/BMPR2^KD complex (Supplementary Fig. 7). Inclusion of the GS domain similarly stabilizes the ALK2^KD N-lobe in the complex (Supplementary Fig. 7a, b), as observed in the analysis of GS-ALK2^KD alone versus ALK2^KD alone (Supplementary Fig. 6). While modest decreases in exchange dynamics are also evident at a few regions peripheral to the C1 dimer interface, these exchange perturbations are generally subtle, indicating that the GS domain does not significantly rearrange the interface (Supplementary Fig. 7c–e). Notably, BMPR2^KD exhibited a lower extent of N-lobe protection upon its heterodimerization with ALK2^KD in solution than with GS-ALK2^KD (Supplementary Fig. 7c, d). This difference was also notable when comparing the dynamics of BMPR2 alone with the ALK2^KD/BMPR2^KD complex (Supplementary Fig. 8c, d). Interestingly, the comparison of the GS-ALK2^KD alone with the

ALK2^KD/BMPR2^KD complex pointed to further stabilization of the GS domain/N-lobe interface of ALK2 in the complex (Supplementary Fig. 8a, b). Collectively, this evidence supports the model in which ALK2 and BMPR2 kinase domains interact primarily through their C-lobes in solution but hint to a potential transient involvement of the N-lobes when the GS domain is present. We will discuss the implications of such potential interactions in Fig. 7.

**Mutations at the C1-dimer interface disrupt ALK2/BMPR2 heterodimerization in solution.** The characteristic feature of the ALK2/BMPR2 C1 kinase dimer interface is the complementarity of the electrostatic surface potentials on both kinases (Fig. 2c). In our structural model of the C1 ALK2/BMPR2 kinase heterodimer, the residues within the acidic patch present on the BMPR2 kinase C-lobe make numerous contacts with the residues in the basic patch in the C-lobe of the ALK2 kinase (Fig. 4a). In agreement with electrostatic interactions being the driving force behind heterodimerization, the recombinant ALK2^KD/BMPR2^KD complex gradually dissociates under conditions of increasing ionic strength (Fig. 4b).

Using the ALK2/BMPR2 C1 kinase dimer model and the HDX-MS measurements as guidelines, we designed mutations to disrupt ALK2^KD dimerization with BMPR2^KD and tested their effect by SEC analysis (Fig. 4c). Due to the broad distribution of the charged residues on both sides of the heterodimer interface and possibly additive effect of electrostatic interactions, we initially altered more than one residue at a time. In one ALK2^KD mutant (ALK2^KD-KK), two lysines within helix I (K492 and

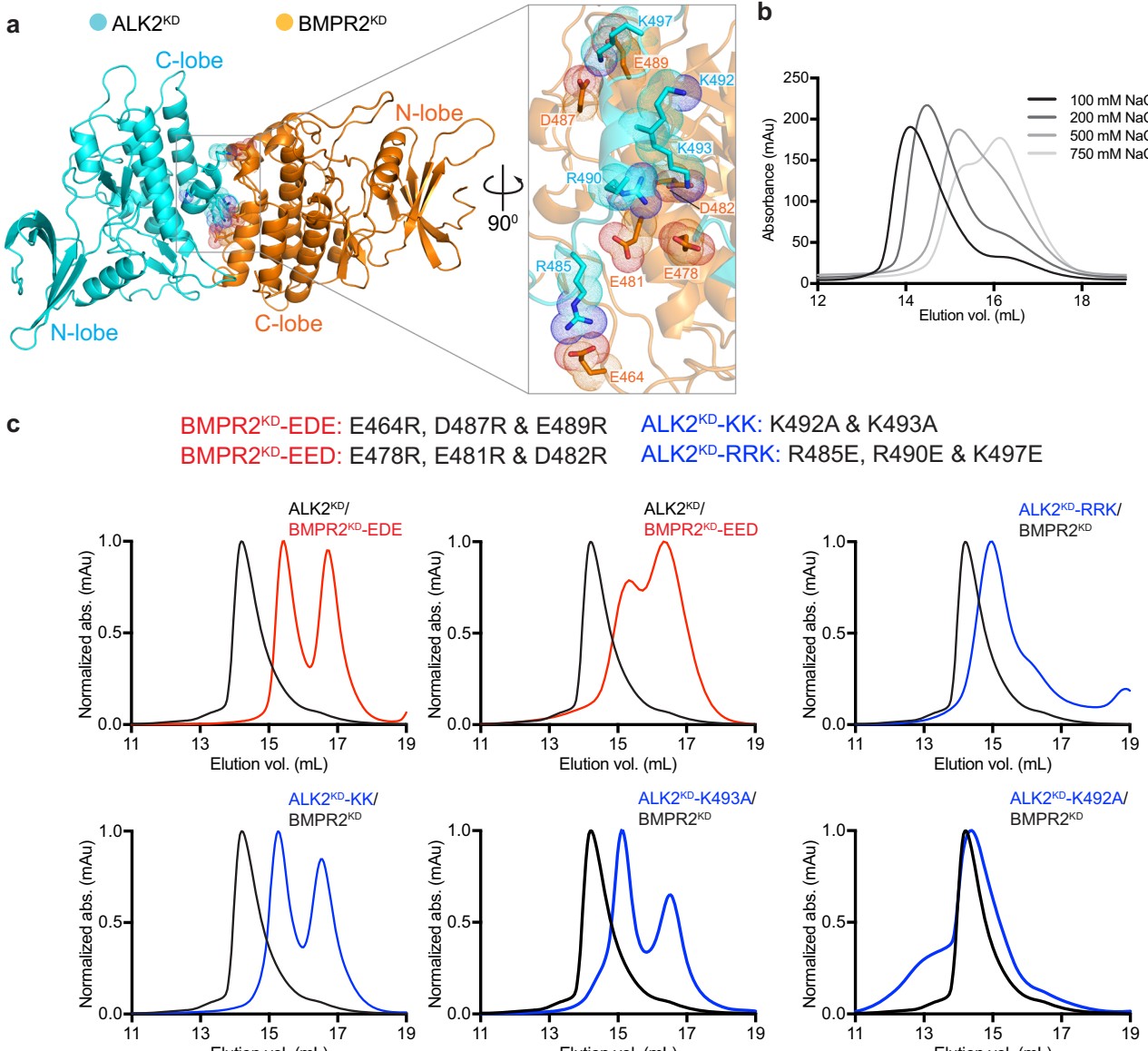

**Fig. 4 Mutation of the C1 dimer interface disrupts the ALK2/BMPR2 heterodimeric kinase complex in solution. a** The C1 dimer with a zoomed in view of the interface showing positions of the charged residues within the C-lobes of the BMPR2 and ALK2 kinases involved in electrostatic interactions. **b** Overlaid size-exclusion chromatograms showing elution of the ALK2$^{KD}$/BMPR2$^{KD}$ complex equilibrated in buffers containing the indicated increasing NaCl concentrations. **c** Nomenclature of the mutations introduced into the ALK2$^{KD}$ and BMPR2$^{KD}$ constructs to investigate the importance of the C1 dimer interface for ALK2$^{KD}$/BMPR2$^{KD}$ complex formation in solution is shown at the top. Bottom panels show size-exclusion chromatograms of the ALK2$^{KD}$/BMPR2$^{KD}$ complexes carrying the indicated mutations (red traces for BMPR2$^{KD}$ mutations and blue traces for ALK2$^{KD}$ mutations) overlaid on the chromatogram obtained for the wild-type ALK2$^{KD}$/BMPR2$^{KD}$ complex.

K493) were mutated to alanines; in another (ALK2$^{KD}$-RRK), R490 in helix I, along with R485 and K497 on adjacent loops, were mutated to glutamates. Both ALK2$^{KD}$ mutants had greatly impaired ability to dimerize with wild-type BMPR2$^{KD}$. In particular, ALK2$^{KD}$-KK completely lost the ability to dimerize (Fig. 4c). In our C1-dimer model, K493 is involved in the interaction with BMPR2, whereas K492 is rotated away from the center of the dimerization interface. The effects of single-point mutations of K493 and K492 are consistent with the more detailed aspects of our model: ALK2$^{KD}$-K493A can no longer dimerize with BMPR2, whereas K492A mutation does not have much effect (Fig. 4c). Hence, K493 emerges as an interaction "hot-spot" on the ALK2 side of the dimer interface.

The BMPR2 residues that directly engage with helix I of the ALK2 kinase in our structural C1-dimer model include E478,

E481, and D482, located on helix H in the BMPR2 kinase. Their mutation to arginines (BMPR2$^{KD}$-EED) almost completely eliminated the ability of BMPR2$^{KD}$ to dimerize with ALK2$^{KD}$ (Fig. 4c). We observed an even stronger effect with another triple BMPR2 mutant (BMPR2$^{KD}$-EDE), in which the residues in the αH-αI linker (D487 and E489) and in the loop N-terminal to helix G (E464) were mutated to arginines. These mutations completely prevented heterodimerization of BMPR2$^{KD}$ with ALK2$^{KD}$ (Fig. 4c). These data corroborate the results of the MD simulations and the HDX-MS measurements, which identify the αH-αI linker in the BMPR2 kinase as a key region within the dimer interface (Fig. 3b). Collectively, the mutagenesis data demonstrate that the ALK2$^{KD}$/BMPR2$^{KD}$ dimer, which forms in solution, has an interface consistent with that of our C1 dimer model.

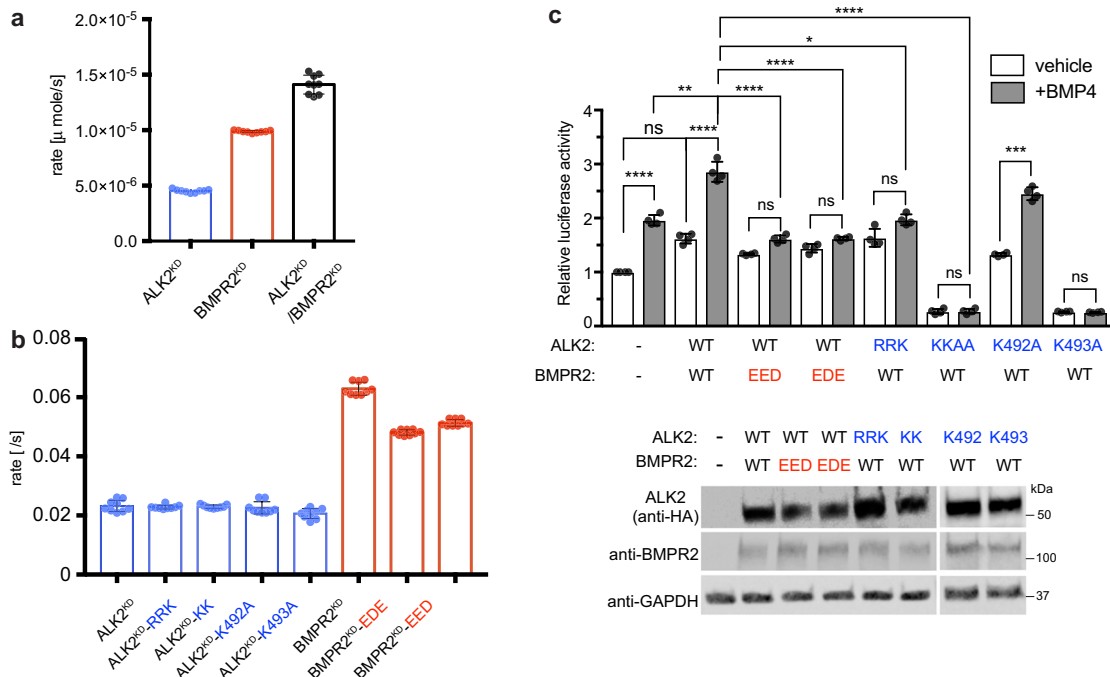

**Fig. 5 Mutation of the C1 dimer interface does not affect kinase activity but prevents ligand-induced signaling by the ALK2/BMPR2 receptor complex in cells. a** In vitro measurement of the kinase activity of the ALK2[KD] and BMPR2[KD] constructs alone or in complex with each other at 4 µM total enzyme concentration. The specific activities of the respective kinase mix in solution are shown. Each kinase assay measurement consists of three independent runs each time of samples in triplicates performed on samples obtained from separate purifications. Data shown are the means with the standard deviation of the means (SDs). **b** In vitro measurement of the kinase activity of the while type ALK2[KD] and BMPR2[KD] constructs and their respective mutant variants carrying substitutions at the C1 dimer interface residues (marked in red for the BMPR2[KD] mutations and in blue for the ALK2[KD] mutations). The rates measured at 2 µM enzyme concentration for each construct are shown. Each kinase assay measurement consists of three independent runs each time of samples in triplicates performed on samples obtained from separate purifications. Data shown are the means with the standard deviation of the means (SDs). **c** Effect of the C1 dimer interface-disrupting mutations on the BMP4-mediated activation of SMAD-dependent transcription by the BMPR2/ALK2 receptor complex. HCT116 cells were transfected with SMAD luciferase reporter construct (SBE-Luc), wild-type or mutant HA-tagged ALK2 with or without wild-type BMPR2, followed by 300 pM BMP4 treatment for 24 h and analyzed by dual luciferase assay. Firefly luciferase activity was normalized to Renilla luciferase activity. Experiments were repeated at least three times and in each experiment each condition was quadruplicated with four independent biological samples. Means and standard deviations of the means (SDs) are indicated. *$P < 0.05$, **$P < 0.01$, ***$P < 0.001$, ****$P < 0.0001$, ns, not significant by ANOVA with two-sided post hoc Tukey's test. The relative expression levels of the transfected constructs in cells were assessed by Western blotting with indicated antibodies (anti-HA antibody was used to detect HA-tagged ALK2).

**C-lobe-mediated dimerization of ALK2 and BMPR2 is dispensable for their catalytic activation but essential for signaling by the ALK2/BMPR2 complex in cells**. The C-lobe-mediated ALK2/BMPR2 kinase dimerization interface is distal from the active sites in both kinases, arguing against its role in enzymatic activation of both kinases. We compared the enzymatic activity of recombinant ALK2[KD] alone, BMPR[KD] alone, or mixing ALK2[KD] and BMPR2[KD] by measuring their autophosphorylation rates in vitro. ALK2[KD] and BMPR2[KD] alone are active in solution, with BMPR2[KD] showing notably higher activity (Fig. 5a). Mixing of the two kinases together at 2 µM, predicted to promote dimerization based on SEC measurements, resulted only in an additive increase in activity demonstrating an insignificant role of the C-lobe-mediated dimer on kinase catalysis (Fig. 5a). This conclusion is further reinforced by the observation that mutations that disrupt ALK2[KD]/BMPR2[KD] heterodimerization in solution, including the ALK2 mutants: ALK2[KD]-KK, ALK2[KD]-RRK, ALK2[KD]-K492A, ALK2[KD]-K493A, and BMPR2 mutants: BMPR2[KD]-EDE and BMPR2[KD]-EED had no effect on the activity of these kinases (Fig. 5b).

To determine whether the C-lobe-mediated dimerization of ALK2 and BMPR2 kinases plays a role in downstream SMAD signaling, we evaluated the effect of the dimer interface-disrupting mutations on the ligand-mediated activation of SMAD-dependent transcription. HCT116 cells were transfected

with a SMAD luciferase reporter construct (SBE-Luc), wild type or mutant full-length ALK2 with or without wild type or mutant full-length BMPR2, followed by BMP4 treatment and the luciferase assay. Co-transfection of wild-type ALK2 and BMPR2 yields a significant increase of BMP4-dependent signals compared to the untransfected control (Fig. 5c). This increase is almost completely eliminated by all mutations at the dimer interface: in ALK2 (ALK2-RRK and ALK2-KK) and BMPR2 (BMPR2-EED, BMPR2-EDE), suggesting that dissociation of C-lobe-mediated ALK2[KD]/BMPR2[KD] dimer impairs signaling capacity of the full-length ALK2/BMPR2 receptor complex (Fig. 5c). The ALK2[KD] double mutant (ALK2-KK) displayed a dominant effect on the BMP4-induced transcription, almost completely eliminating the basal reporter activity (Fig. 5c). In agreement with the SEC analysis in Fig. 4c in which K493A mutation, but not K492A, disrupted ALK2/BMPR2 dimerization. The inhibitory effect on signaling was also selective to the mutation of K493, while the mutation of K492 did not have a significant effect upon BMP4 treatment (Fig. 5c). These results underscore a critical role of C-lobe-mediated dimerization between ALK2 and BMPR2 kinases for transmitting the signal to R-SMADs in response to ligand binding.

The strong effect on signaling of the K493A mutation located in helix I of ALK2 prompted us to examine if this mutation

promotes any specific structural changes to the ALK2 kinase domain structure. We crystallized the ALK2KD-K493A mutant and solved its structure at 2.4 Å resolution in complex with ATP analogue, AMPPNP (Supplementary Fig. 9a) (Supplementary Table 2). We also obtained crystal structures of the ALK2KD-KK in the presence of AMPPNP and the ALK2 kinase inhibitor, LDN-193189 (Supplementary Fig. 9b, c) (Supplementary Table 2). These structures of the ALK2 kinase domain with an ATP analog demonstrate that K493A mutation or K492A/K493A double mutant do not alter the ALK2 kinase domain structure when compared to wild-type ALK2. Hence, the disruptive effect of these mutations on BMP4-induced ALK2/BMPR2 signaling underscores a functional role of the C-lobe-mediated ALK2/BMPR2 heterodimer in the formation of an active receptor complex.

**The BMPR2 C-lobe dimer interface is disrupted in PAH.** Heterozygous loss-of-function or expression mutations in *BMPR2* account for ~80% of familial and ~20% of idiopathic PAH[27]. Some of these mutations introduce insertions, cause deletions or frameshifts in the BMPR2 kinase domain, likely compromising its structure and hence enzymatic activity. The mechanisms of action of missense mutations commonly found on the surface of the BMPR2 kinase are more difficult to rationalize[28–31]. Remarkably, several of these mutations (C483R, D485G, D487V, A490D/V, and R491W/Q) cluster to the C1 dimer interface in the BMPR2 kinase domain, specifically at the αH-αI linker (Fig. 6a). As shown in Figs. 4c and 5c, mutations of the residues within the αH-αI linker disrupt ALK2/BMPR2 kinase dimerization and downstream SMAD signaling by the full-length ALK2/BMPR2 receptor complex (Figs. 4c and 5c). Two sites of PAH mutations—D487 and D485—stand out because they make direct interactions with residues in ALK2 in the C1 dimer model (Fig. 6b). D487 was included in the BMPR2KD-EDE mutant which failed to dimerize with ALK2KD (Fig. 4c). D485 was not included in our initial mutagenesis and was of particular interest. The D485G mutation is highly penetrant in PAH, and previous studies have shown its dominant-negative effect on BMPR2 signaling in complex with other type I receptors (ALK3 and ALK6) in cells[30,32,33].

When the D485G mutation was introduced into BMPR2KD (BMPR2KD-D485G), the recombinant BMPR2KD-D485G did not dimerize with the wild-type ALK2KD (Fig. 6c). This mutation also had a dominant-negative effect on BMP4-dependent SMAD signaling by the ALK2/BMPR2KD-D485G full-length receptor complex (Fig. 6d). Due to prior speculations that loss of the acidic charge at residue 485 compromises the kinase fold[19,34], we obtained a crystal structure of the BMPR2KD-D485G kinase solved to 2.3 Å resolution (Fig. 6e) (Supplementary Table 2). The BMPR2KD-D485G structure revealed an intact architecture of the kinase domain and overlaid with the recently reported structure of the wild-type BMPR2 kinase domain[35] with an RMSD of 0.5 Å over 279 Cα atoms. Moreover, like other C-lobe mutants tested (BMPR2-EED and BMPR-EDE), the BMPR2KD-D485G mutant exhibited enzymatic activity comparable to the wild-type BMPR2KD (Fig. 6f). Thus, we conclude that the pathological effect of the D485G mutant of BMPR2 in PAH patients is due to loss of the receptor's ability to engage in C-lobe-mediated heterodimers, which results in loss of downstream signaling.

**A model of the active type I/type II receptor kinase heterotetramer.** In the active heterotetrameric complex composed of two type I and two type II receptors, the type II kinase releases autoinhibition of the type I kinase by phosphorylating the GS domain. In the C-lobe-mediated kinase dimer the N-terminus of the ALK2 kinase, where the GS domain is located, is positioned too far away from the active site of the BMPR2 kinase to enable

phosphorylation of the GS domain. Such sequestration of the GS domain away from the active site of the type II kinase is consistent with receptor autoinhibition but our data show that the C-lobe mediated dimerization plays an activating role in ALK2/BMPR2/SMAD signaling (Fig. 5c). Since the active receptor complex is a tetramer, we therefore hypothesized that the importance of the C-lobe-mediated interface in activation might stem from its involvement in orienting kinases for proper GS domain phosphorylation in the tetramer via another type I/type II kinase interface.

The N dimer identified from our MD simulations has a dimerization interface on the ALK2 and BMPR2 kinase N-lobes, both of which remain exposed when the two kinases interact by way of their C-lobes (Fig. 2a). The formation of the ALK2/BMPR2 kinase dimer using these interfaces (the N dimer) would displace the GS domain from the N-lobe of ALK2, a necessary step toward subsequent phosphorylation by the BMPR2 kinase (Supplementary Fig. 3a). To investigate if the C-lobe-mediated dimer provides a plausible scaffold for the accommodation of the ALK2/BMPR2 N dimers in a kinase tetramer, we first constructed a model in which the ALK2 from one N dimer (ALK2′) and BMPR2 from the second N dimer (BMPR2″) were overlaid onto the structure of the ALK2/BMPR2 C1 dimer to produce an initial tetramer model. In this tetramer, the C-lobes of the remaining two kinases (BMPR2′ and ALK2″) were directed away from each other, preventing the formation of a closed symmetric tetrameric complex (Supplementary Fig. 10a).

As already noted, our MD simulations identified a number of other orientations adopted by the ALK2 and BMPR2 kinase C-lobes that engaged the complementary electrostatic interfaces on both kinases. Overlaying two ALK2/BMPR2 N dimers onto another C-lobe/C-lobe dimer identified in a cluster of these interactions produced a closed and symmetric heterotetramer (Fig. 7a) (see Supplementary Data 2 for the structural coordinates). In this tetramer, both ALK2′/BMPR2″ and ALK2″/BMPR2′ C-lobe-mediated interfaces are the same (Supplementary Fig. 10b). Compared to the C1 dimer, the new interface (which we call the C2 dimer interface) forms when ALK2 kinase is rotated with respect to BMPR2 kinase, bringing helices H and G of ALK2 into contact with helices E and G of BMPR2. The electrostatic complementarity at the interface, however, remains intact. Although the C2 dimer is less compatible with the HDX-MS data overall than the C1 dimer, it does engage the key residues that form the electrostatic interactions that we have validated experimentally as being important for SMAD signaling, including the hot-spot residues: K493 on ALK2 and D485 on BMPR2 (Supplementary Fig. 10c). Indeed, at the protein concentrations used for HDX-MS analysis (6 μM) the kinase domains may have formed a mixed population of C1 dimer and C2-mediated heterotetramer.

The architecture of the ALK2/BMPR2 tetramer mirrors the symmetry that is seen in the structures of the type I/type II tetrameric complexes of the extracellular domains (Supplementary Fig. 11). In the kinase tetramer, the N-terminal ends of both ALK2 kinases are presented toward the active sites of the BMPR2 kinases within their respective N dimers, but across two different C-lobe dimer interfaces (Fig. 7b). Using a structure of the PKA kinase bound to a substrate peptide[36], we created a model of the interaction between the GS-domain peptide extending from the N-lobe of ALK2 toward the active site of the BMPR2 kinase. This tetramer pose was unchanged over the long time-course of the MD simulations (Supplementary Fig. 10d). Thus, the C-lobe- and N-lobe-mediated interfaces support the formation of a tetrameric complex in which all phospho-sites in the ALK2 GS domain are presented for phosphorylation. Another feature of the tetramer model consistent with it representing the active receptor complex is that it precludes binding between ALK2 and the negative regulator FKBP12, due to

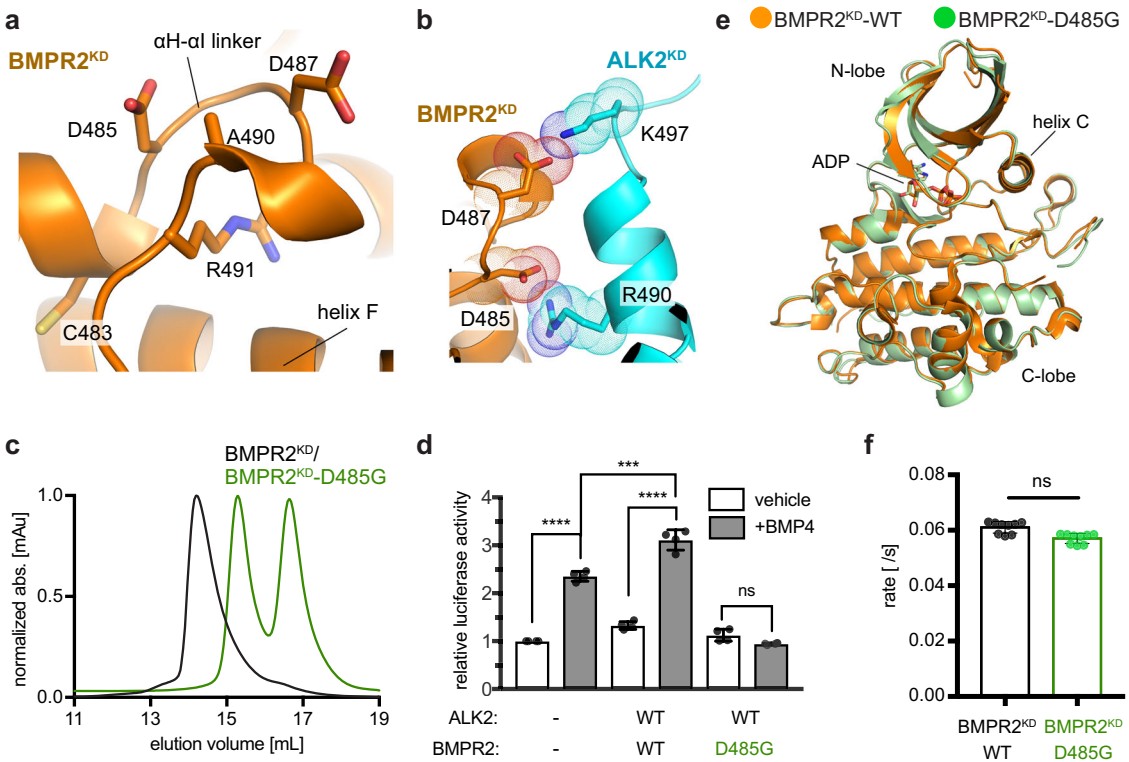

**Fig. 6 Several PAH mutations map to the C1 dimer interface in the BMPR2 kinase and block its dimerization with the ALK2 kinase. a** Zoomed in view of the αH-αI linker within the C1 dimer interface in the BMPR2 kinase depicting residues C483, D485, D487, A490, and R491W mutated in PAH patients. **b** Zoomed in view depicting interactions made by D485 and D487 in BMPR2 with basic side chains of the residues at the C1 dimer interface in the ALK2 kinase. **c** Size-exclusion chromatograms of the ALK2$^{KD}$/BMPR2$^{KD}$-D485G complex (green trace) overlaid on the wild-type ALK2$^{KD}$/BMPR2$^{KD}$ complex (black trace). **d** Effect of the D485G mutation on the BMP4-mediated activation of SMAD-dependent transcription by the BMPR2/ALK2 receptor complex, measured as described in Fig. 5c. Experiments were repeated at least three times and in each experiment each condition was quadruplicated with four independent biological samples. Means and standard deviation of the means (SDs) are indicated. $*P < 0.05$, $**P < 0.01$, $***P < 0.001$, $****P < 0.0001$, ns, not significant by ANOVA with two-sided post hoc Tukey's test. **e** A crystal structure of the BMPR2$^{KD}$-D485G kinase, shown in green overlaid on the crystal structure of the wild-type BMPR2 kinase domain shown in orange (PDB: 3G2F). **f** In vitro measurement of the kinase activity of the wild-type BMPR2$^{KD}$ and BMPR2$^{KD}$-D485G constructs. The rates measured at 2 µM enzyme concentration for each construct are shown. Each kinase assay measurement consists of three independent runs each time of samples in triplicates performed on samples obtained from separate purifications. Data shown are the means and standard deviation of the means (SDs) and was calculated using two-sided $t$-test, with no adjustments for multiple comparisons (ns, not significant, $P > 0.05$).

engagement of the N dimer interface (Supplementary Fig. 3a). The tetramer model thus provides a mechanism for FKBP12 release from ALK2 during ligand-induced receptor activation and subsequent GS domain phosphorylation.

While the tetramer was structurally stable in simulations, the constituent C2 dimer within a tetramer was not stable by itself in contrast to the C1 and N dimers (Supplementary Figs. 10e and 12). This suggests that the BMPR2-ALK2 C-lobe/C-lobe interaction in the tetramer is strained and is primarily stabilized by the N-lobe-driven interactions, which were stable in the MD simulations (Fig. 2b, Supplementary Fig. 12). This might explain why we do not see tetrameric complexes formed by the isolated kinase domains in solution and suggests that the tetramer only forms when the N-lobe-mediated interactions between type I and type II kinases are enabled, likely after the extracellular domain tetramer is stabilized by ligand binding. Other interactions might also contribute to the stabilization of the kinase tetramer structure, such as the binding of ALK2 GS-domain in the active site of the BMPR2 kinase (Fig. 7b). In support of this speculation is our observation that in the presence of the GS domain, there are HDX perturbations in the N-terminal lobes of both kinases in the ALK2/BMPR2 complex (Supplementary Figs. 7 and 8) and that these signals are consistent with the N-lobe driven interactions in the kinase tetramer (Supplementary Fig. 13).

The N-dimer interface between the ALK2 and BMPR2 kinases is symmetric and encompasses surface-exposed residues on helix C of both kinases, including a central patch of hydrophobic residues (ALK2: W245 and F246; BMPR2: F240 and I241) with complementary electrostatic pairings at either end of the helix (Fig. 7c). We mutated the hydrophobic residues in the center of this interface (F246R in ALK2 and I241E in BMPR2) to test their effect on SMAD signaling in response to ligand binding. N-lobe mutations in ALK2 that interfere with the GS domain binding have an activating effect on signaling[17,18], but the F246R mutation is peripheral to the GS domain-binding interface in the N-lobe of ALK2 and it should be specific for the tetramer interface. Both ALK2-F246R and BMPR2-I241E mutations markedly impaired SMAD signaling in response to BMP4 stimulation, supporting the functional significance of the proposed tetramer model for receptor signaling in the context of the ligand-bound full-length type I/type II receptor complex in the cell (Fig. 7c).

**Conservation of the oligomerization interfaces among the TGFβ receptor kinases.** Structural studies on the extracellular domains of different members of the TGFβ superfamily reveal the

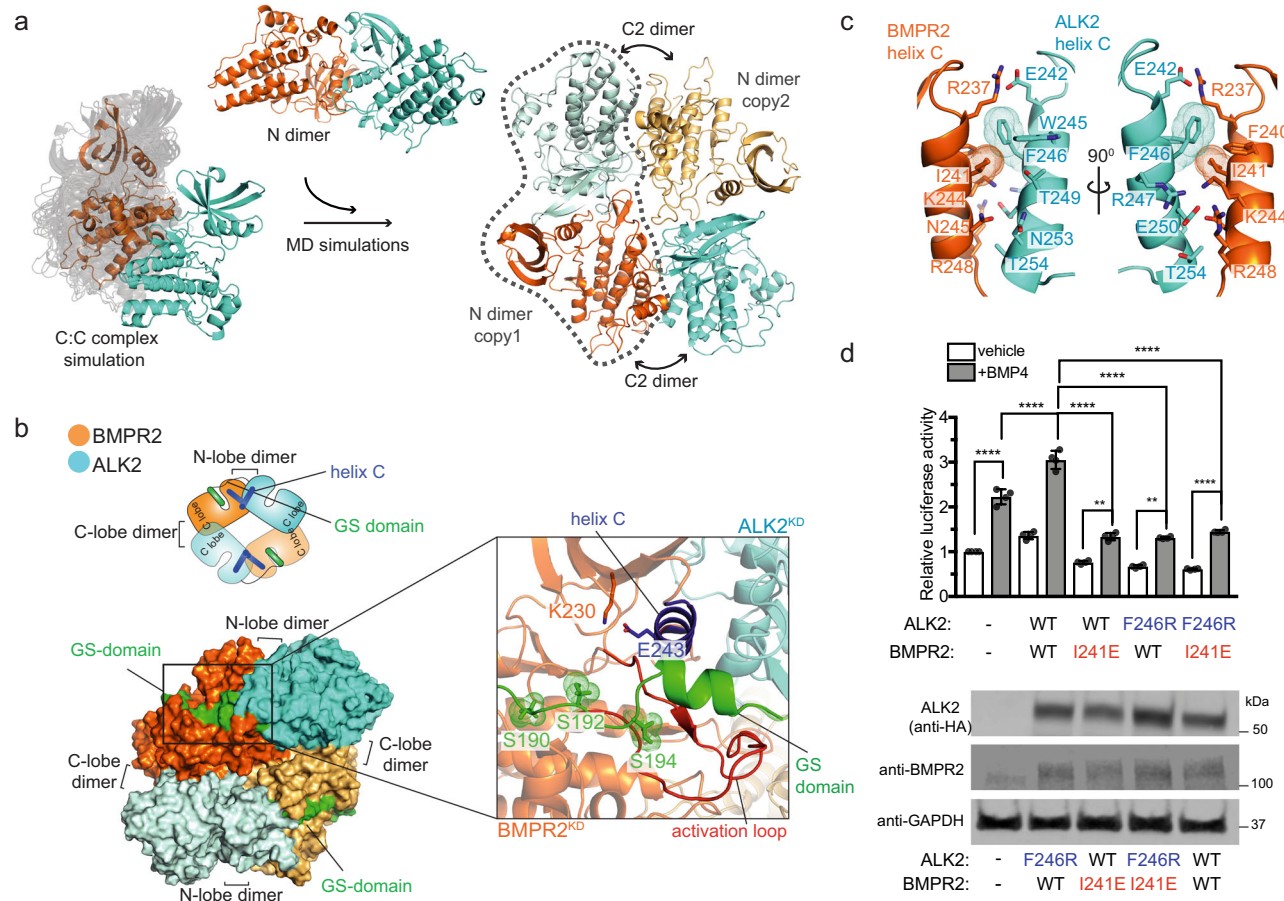

**Fig. 7 The C1 dimer interface provides a scaffold for the association of the ALK2/BMPR2 kinase tetramer via the N dimer interface. a** Two identical copies of the N dimer were superposed on the trajectory of an MD simulation of the C-lobe/C-lobe 1 dimer where we observed slight variations of the C-C interface. A pose that put both N dimer copies at a close distance was extracted and was subjected to more MD simulations (50 μs aggregate simulation time described in Supplementary Table 5) to generate the tetramer model where tetramer interfaces converged to a consensus pose. **b** The interaction between the GS domain of ALK2 and the active site of the BMPR2 kinase domain is modeled in both ALK2 monomers in the tetramer using a crystal structure of PKA (PDB: 1ATP). This interaction occurs across the N dimer interfaces and positions the GS domain for phosphorylation. The inset on the right zooms in on the interaction between the GS domain and the BMPR2 kinase active site. The phosphorylatable serine residues in the GS domain are shown in dot representation. **c** Zoomed in view of the N dimer interface between the ALK2 and BMPR2 kinases in the ALK2/BMPR2 tetramer model highlighting key residues within helices C of each kinase that contributes to the interface. **d** Effect of the N dimer interface mutants on the BMP4-mediated activation of SMAD-dependent transcription by the BMPR2/ALK2 receptor complex, measured as described in Fig. 5c. Experiments were repeated at least three times and in each experiment each condition was quadruplicated with four independent biological samples. Means and standard deviations of the means (SDs) are indicated. *P < 0.05, **P < 0.01, ***P < 0.001, ****P < 0.0001, ns, not significant by ANOVA with two-sided post hoc Tukey's test. Western blot analysis of expression levels of ALK2 and BMPR2 receptors carrying the N dimer interface mutations expressed transiently in HEK293 cells upon cell lysis is shown in the bottom panel.

universal tetrameric architecture of their ligand-induced complexes[8]. The tetrameric state (2 type I/2 type II) is preserved despite structural differences that exist within the extracellular domain tetramers, especially between features that distinguish BMP and TGFβ receptors[4]. While future studies will reveal whether the described ALK2/BMPR2 C-lobe-mediated dimer and its further assembly into a tetramer are compatible with other receptor pairs, analysis of the relevant kinase C-lobe interfaces in other receptors, whose structures have been solved thus far, demonstrates universal conservation of the positive electrostatic surface potential among the type I receptors and a corresponding negative surface in the type II receptors (Fig. 8a). Notably, the complementary electrostatics are maintained despite differences in the conservation of individual residues at the opposing interfaces (Fig. 8b).

The N-lobe-mediated dimer interface is significantly conserved as well, which might in part reflect its encapsulation of helix C, and, in the type I TGFβ superfamily of receptors, the conserved function

of this interface in binding of the GS domain. Consequently, among the type I receptors, the N-lobe-mediated interface is conserved to a much higher degree than in type II receptors, with the exception of two closely related type II activin receptors, (ACVR2a and ACVR2b) (Supplementary Fig. 14). However, the key features of the N-lobe-mediated dimer interface in type II receptors which are important for tetramer formation in our model are represented in most receptors. All type II receptors have a hydrophobic residue in the same location as F240 in BMPR2, and with the exception of ACVR2a, a basic residue corresponding to K244 in BMPR2, which can engage with a conserved acidic residue of the type I receptor (E250 in ALK2) (Fig. 7c).

**Prevalence of disease mutations at the C-lobe-mediated dimer interface among the TGFβ superfamily of receptors.** The importance of the C-lobe dimerization interface is further

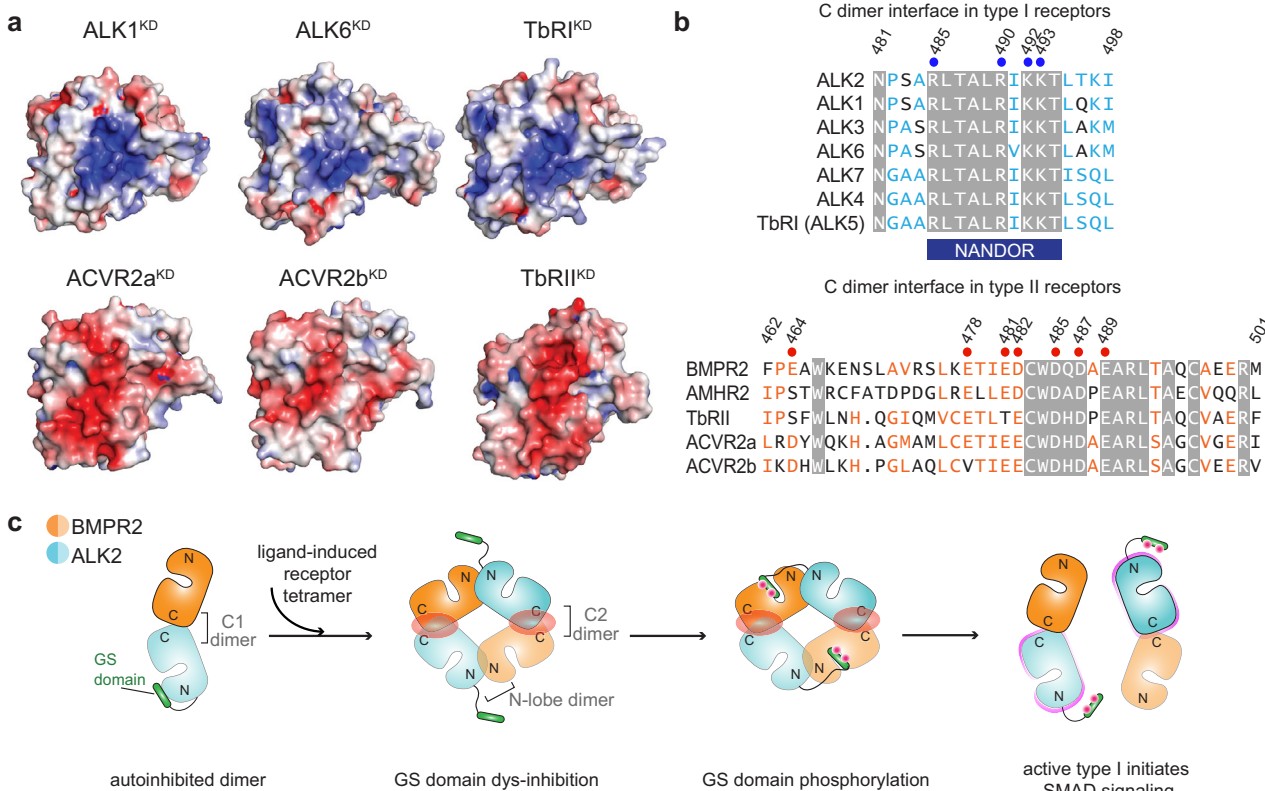

**Fig. 8 Conservation of the electrostatic C-lobe interfaces and the N dimer interface in the TGFβ superfamily receptors. a** The electrostatic surface potential at the C1 dimer interface calculated using APBS and the available crystal structures of the TGFβ superfamily of kinases. Type I receptors: ALK1 (PDB: 3MY0), ALK6 (PDB: 3MDY) and TGFβR1 (PDB: 5E8S). Type II receptors: ACVR2a (PDB: 3Q4T), ACVR2b (PDB: 2QLU), and TGFβR2 (PDB: 5E8V). All type I kinases are oriented the same way as the ALK2 kinase and type II kinases as the BMPR2 kinase in Fig. 1c. **b** Sequence alignment of the regions within the human TGFβ superfamily of receptors that correspond to the C dimer interface. Identical residues are highlighted by a gray background. Similar residues are colored as light blue in type I and light orange in type II receptors. Residues tested by mutations in our experiments are marked by circles and numbered. The position of the TGFβR1 NANDOR (non-activating non-down-regulating) box is highlighted in dark blue below the sequence alignment. **c** The proposed model for the oligomeric transitions between the ALK2 and BMPR2 kinase domains in response to ligand binding. The interaction of ALK2 and BMPR2 in the C1 dimer provides an autoinhibitory mechanism in which the GS domain of ALK2 is protected from BMPR2-mediated phosphorylation. Upon ligand-induced formation of the receptor tetramer, the kinase domains are brought into close proximity promoting interactions between ALK2 and BMPR2 via the N dimer, and the presentation of the GS domains to the active sites of the BMPR2 kinases for phosphorylation. When the GS domains are phosphorylated, the tetramer could dissociate to two C1 dimers in which the ALK2 kinases are now active, cartooned as a magenta rim around them.

underscored by the accumulation of disease mutations in this region in the TGFβ superfamily of receptors (Supplementary Fig. 15a). We show here that several PAH mutations directly disrupt the BMPR2 C-lobe interface and prevent its interaction with ALK2, resulting in loss of receptor signaling. Among more than 300 mutations cumulatively detected in the BMPR2 gene in familial and idiopathic versions of PAH, many are located in the kinase domain, primarily in the C-lobe[37]. While mutations that are buried within the kinase C-lobe likely inhibit BMPR2 signaling by destabilizing the structure of the BMPR2 kinase[35], the mechanism of action of mutations localized on the surface of the kinase C-lobe has been a long-standing puzzle in the field. Thus, our data provide the first mechanistic explanation for the loss-of-function phenotype of this type of BMPR2 mutation. Notably, the type II TGFβ receptor (TbRII), another member of the TGFβ receptor superfamily, also accumulates disease-related mutations in its C-terminal kinase lobe, directly adjacent to the dimerization interface we describe here (Supplementary Fig. 15b). These mutations are implicated in connective tissue disorders whose clinical manifestations overlap to varying extents with those of Marfan syndrome, a disorder characterized by a combination of characteristic vascular, skeletal, and other features[38]. In addition, a TbRII-inactivating mutation (P525L)

was identified in a screen that compromises ability of TbRII to recognize type I receptor ALK5 (TbRI) as a substrate, without affecting TbRII catalytic activity or ligand binding[39]. The mutated residue, P525 in TbRII, is equivalent to A488 in BMPR2, localized in the center of the C-lobe-mediated interface, raising the possibility that the P525L mutation might compromise the ability of the TbRII kinase to engage the type I kinase in a C-lobe-mediated dimer.

Most of the residues within the C-lobe-mediated dimerization interface in ALK2 (including R485, R490, K492, and K493) fall within a highly conserved C-terminal region in the type I TGFβ receptors once denoted as the NANDOR (non-activating non-down-regulating) box (Fig. 8b). As discussed below, this region has been shown to be important for signaling by a number of type I TGFβ receptors and to be a hot spot for mutations in a spectrum of diseases (Supplementary Fig. 15a). In ALK5, deletions or mutations within the NANDOR box eliminate receptor signaling[40]. Inactivating missense mutations within this region of ALK5, R487W/Q/P, have been associated with the Marfan-like Loeys-Dietz syndrome, type 1[41]. The R487 residue in ALK5 is equivalent to R490 in ALK2 and is at the center of the C-lobe-mediated dimerization interface in our ALK2/BMPR2 kinase dimer model (Fig. 4a). Another type I receptor, ALK1, is mutated

in cardiovascular diseases such as hereditary hemorrhagic telangiectasia (HHT) and PAH[42]. In these disorders, several mutations have been detected in the ALK1 kinase domain that fall within the C-lobe-mediated dimerization interface that we describe here, including the R484W/Q mutation (again equivalent to the R490 in ALK2) and K487T (equivalent to K493 in ALK2)[42,43]. The type-I receptor, ALK6, also acquires a pathogenic mutation in the C-lobe interface, R484W, which results in brachydactyly type A2[44]. This residue in ALK6 is equivalent to the R490 in ALK2, and its mutation also results in loss of receptor signaling without affecting its catalytic activity[44]. Collectively, these data emphasize the importance of the C-lobe interface in type I kinases and support the hypothesis that the C-lobe interface dimerization mechanism is shared across the TGFβ receptor superfamily.

## Discussion

Assembly of the tetrameric complex between two type I and two type II receptors is believed to be at the core of activation of all members of the TGFβ receptor superfamily and points to the critical role of a tetramer in the activation mechanism. However, since no structures of the complexes formed by the intracellular portions of these receptors exist, the need for receptor tetramerization for kinase signaling has been a long-standing mystery. Here, using an integrative structural biology approach to study the interaction between the type I kinase ALK2 and the type II kinase BMPR2, we reveal their heterodimeric assembly via complementary electrostatic C-lobe-centered interfaces, which is essential for the signaling of this receptor complex. We also postulate that further assembly of these heterodimeric complexes into a tetramer is needed to enable phosphorylation of the autoinhibitory GS domain, a critical step in type I/type II receptor complex activation. Thus, we propose the first structural model for the assembly of the active kinase complex that provides a mechanistic explanation for how tetramerization leads to the activation of the type I receptor kinases.

The C-lobe-mediated interaction between the BMPR2 and ALK2 kinases sets an efficient mechanism for receptor autoinhibition in the absence of ligand binding because it positions the GS domain of ALK2 a safe distance away from the active site of BMPR2, preventing GS domain phosphorylation and subsequent ALK2 activation. Ligand-independent heteromeric assemblies of type I and type II receptors have been observed and are particularly frequent for the BMP receptors[45], estimated to engage 20% of all receptors[23]. These preformed dimers are hypothesized to play an important role in receptor activation via the so called preformed complex (PFC) activation mode[46]. As noted before[3], such complexes need to remain inactive until ligand is bound, invoking an autoinhibitory mechanism that efficiently prevents undesired phosphorylation of the GS domain in the absence of a signal. The C-lobe-mediated interface we describe presents such an autoinhibitory mechanism.

Tight regulation of basal activity is well characterized in close cousins of TGFβ receptors—RTKs[47]. In RTKs, autoinhibition has been described to span a variety of mechanisms—from the formation of inactive dimers in InsR to allosteric inhibition by the juxtamembrane domain as observed in Eph receptors[48] and the PDGF family of receptors[49]. Like in TGFβ receptors, juxtamembrane domain-mediated inhibition can be removed via its phosphorylation upon ligand-induced oligomerization of RTKs[47]. There are almost no insights into the structural mechanisms by which these phosphorylation events occur in RTKs, and they potentially might involve other kinases so the autoinhibitory locks set up by the juxtamembrane domains on the receptor kinase domains can be released. Here we show that the tetrameric architecture of the type I/type II complex provides a suitable scaffold to support autonomous phosphorylation of the GS domain-containing juxtamembrane domain within the receptor complex.

While the C-lobe-mediated dimer plays an essential role in autoinhibition of the type I/type II heterodimer, it also seems critical for the assembly of the active receptor tetramer. The C-lobe-mediated kinase interactions keep the N-lobes of the type I and type II kinases available to engage, enabling assembly of the tetrameric complex in which two C-lobe dimers interact via the N dimer interface. Such a kinase tetramer mirrors the architecture of the extracellular domain tetramer in which type I and type II receptors alternate positions, and the N-lobe-mediated interaction favorably positions the kinases to facilitate the displacement of the inhibitory domains away from the type I kinase and to position the GS domain into the active site of the type II kinase. While the tetrameric complex does not form stably in solution, our HDX analysis hints to the possible N-lobe engagement, which is further accentuated by the presence of the GS domains. In the context of a ligand-stabilized tetrameric complex of full-length receptors, apparent affinity of N dimer interactions is predicted to significantly increase, facilitating GS domain binding to the active sites of type II kinase as delineated in our tetramer model.

We also show that to form a tetramer, the C1 dimer interface between the ALK2 and BMPR2 kinases needs to rotate slightly along the complementary electrostatic surfaces of the type I and type II kinases. We posit that after the GS-domain phosphorylation and release from the BMPR2 active site, the kinase tetramer structure will become unstable and break into two C1 dimers again (Fig. 8c). These two C1 dimers are now catalytically active—with the GS domains released from the autoinhibitory position due to their phosphorylation—and can serve as the basic signaling units for recruitment of SMAD substrates. Mutations in the C-lobe dimer interface are poised to break these scaffolding capabilities at every step of the activation process, and thus their negative effect on SMAD signaling that we observe is consistent with the proposed tetramer model. Prevalence of disease mutations that map to the C-lobe dimer interface across many receptors in the TGFβ family further underscores the critical role of this interface in proper receptor signaling.

The postulated plasticity of the C-lobe-mediated dimer interface has important implications when considering the conservation of the described activation mechanism across other receptors in the TGFβ receptor superfamily. While the ALK2/BMPR2 complex is not the most common receptor pairing between the type I/type II receptors in the TGFβ superfamily, we show that the respective surface electrostatic potentials at the C-lobe dimer interface, positive for type I receptors and negative for type II receptors, are conserved. Importantly, while most of the electrostatic side chains are also conserved, other residues at the interfaces show variation. This suggests that any type I or type II receptor has the potential to engage in the electrostatic C-lobe-mediated type I/type II heterodimer, but there will likely be variation in structure, dynamics, and thermodynamic stability between receptor dimers of different combinations. Hence, the described C-lobe-mediated heterodimerization might be a universal mode for assembly of active receptor complexes in the TGFβ superfamily and could explain how these structurally diverse receptors evolved to follow the same principle for activation within a tetramer and engage in promiscuous interactions with each other. Any disease mutations that target the C-lobe interfaces carry the potential to disrupt an entire ensemble of possible interactions between the given receptor and its dimerization partners.

In summary, we characterize and functionally validate the first structural model of a type I/type II kinase heterodimer with features consistent with an autoinhibited complex that is preassembled for efficient oligomerization into a tetramer upon ligand binding. This tetrameric kinase assembly might play a role beyond type I kinase activation—for example in effective recruitment of downstream effectors such as SMADs, as suggested by studies showing that constitutively activate ALK2 still requires the type-II receptor kinase domain for signaling[19]. While it remains to be shown if the described tetramer model applies to other members of the TGFβ receptor superfamily, the location of numerous disease mutations at the proposed oligomer interfaces in receptors other than BMPR2 and ALK2, strongly suggests such a possibility. Lastly, this work provides a new platform for the design of BMP receptor-targeted therapeutics. While these efforts have predominantly focused on small-molecule inhibitors of type I receptor kinases to target proliferative disorders, such as FOP and cancer, many BMP receptor-driven disorders are due to loss-of-function of these receptors. Thus far, activation of mutant receptors has been achieved either through ligand stimulation[50,51], FK506-induced inhibition of FKBP12[52,53], or novel small molecule upregulators[54]. The protein–protein interfaces that we identified in this study provide opportunity for the design of molecules that thermodynamically stabilize active receptor states as a potential therapy for PAH and HHT.

## Methods

**Cloning of expression constructs**. The full-length type I BMP receptor, ALK2 (Uniprot ID Q04771, residues 1–509), was previously cloned into pcDNA3.1 with a C-terminal HA-tag[13]. The full-length type-II BMP receptor, BMPR2 (Uniprot ID Q13873, residues 1–1038), was previously cloned into pcDEF3 with a C-terminal FLAG-tag[55]. The kinase domain of ALK2 alone (ALK2$^{KD}$, residues 201–499), or with the GS-domain (GS-ALK2$^{KD}$, residues 172–499), were cloned into the pFastBac insect cell expression vector (Thermo Fisher Scientific) with an N-terminal HIS-tag and a TEV cleavage site. The kinase domain of BMPR2 (BMPR2$^{KD}$, residues 189–529), was cloned into the pET28a bacterial expression vector (Millipore/Sigma) with an N-terminal His-tag and TEV cleavage site. Mutations were engineered with overlapping primers using standard QuikChange mutagenesis protocol (Stratagene). A complete list of primers used for cloning of these constructs is provided in Supplementary Table 3. For ALK2 mutant mammalian expression plasmids: The *BbsI-BglII* fragment in the wild-type ALK2 construct was replaced with the *BbsI-BglII* fragment in the mutant ALK2 construct. The replaced fragment was validated by sequencing For BMPR2 mutant mammalian expression plasmids: The *BspHI-Bpu10I* fragment in the wild-type BMPR2 construct was replaced with the *BspHI-Bpu10I* fragment in the mutant BMPR2 construct. The replaced fragment was validated by sequencing.

**Protein expression and purification**. The kinase domains of ALK2 and BMPR2 were expressed based on previously published protocols, which are described in detail next[35,56]. Recombinant ALK2 kinase domain constructs were expressed in SF9 insect cells. Eight 1 L cultures of SF9 insect cells ($2 \times 10^6$ cells/mL) were infected with 25 ml/l of P2 virus and incubated for 72 h at 27 °C, 110 rpm. Cells were collected by centrifugation and the pellets resuspended in ice-cold lysis buffer (50 mM HEPES pH 8.0; 500 mM NaCl, 5% glycerol, 0.5 mM TCEP, and 10 mM Imidazole) supplemented with 1 EDTA-free protease inhibitor tablets per 100 mL (Roche). The cells were lysed with an Emulsiflex C5 homogeniser and the insoluble fraction removed by centrifugation at $40,000 \times g$ for 1 h. Ni-NTA resin (0.5 mL of 50% slurry per L of insect cells; ThermoFisher Scientific) was pre-equilibrated in binding buffer (50 mM HEPES pH 8.0, 300 mM NaCl, 0.1 mM TCEP, and 10 mM Imidazole) and incubated with the supernatant for 1 h at 4 °C on a roller. Weakly bound proteins were cleared by repeated washes in buffer with 20 mM imidazole. Recombinant kinase domain was eluted with buffer containing 50, 100, and 250 mM imidazole. Protein-containing fractions were assessed by SDS-PAGE, pooled, and 1 mg of TEV protease was added. The sample was incubated at 4 °C for 16 h with very gentle agitation, on an orbital shaker, to avoid precipitation. The buffer was exchanged by centrifugal concentration to 50 mM HEPES pH 8.0, 300 mM NaCl, 0.1 mM TCEP, and 10 mM Imidazole and the uncleaved protein was removed by passage over pre-equilibrated Ni-NTA resin. A final purification was performed by SEC using a Superdex 200 16/60 column equilibrated in 50 mM HEPES pH 8.0, 300 mM NaCl and 0.1 mM TCEP. Recombinant ALK2 kinase domain was concentrated to ~20 mg/ml in 50 mM HEPES pH 8.0, 100 mM NaCl and 0.1 mM TCEP, flash frozen in liquid nitrogen and store at −80 °C.

The kinase domain constructs of BMPR2 were expressed in BL21 *E. coli*. TB media, supplemented with 50 μg/ml kanamycin, was inoculated with 5 ml/l of overnight culture, and incubated at 37 °C and 200 rpm. Upon cell density reaching

OD$_{600}$ = 0.6–1.0 expression was induced with 0.5 mM IPTG and the cells then incubated for 16 h at 18 °C and 200 rpm. The cells were pelleted by centrifugation, resuspended in binding buffer with 5 mM imidazole and 1 EDTA-free protease inhibitor tablet per 100 mL (Roche), and lysed with an Emulsiflex C5 homogeniser. After binding to equilibrated Ni-NTA resin (1 mL of 50% slurry per L of culture; ThermoFisher Scientific), the recombinant kinase domains were eluted with 50 and 100 mM imidazole and the HIS-tag was cleaved overnight with TEV protease. The proteins were then loaded onto a Superdex 200 16/60 column equilibrated in 50 mM HEPES pH 8.0, 300 mM NaCl and 0.1 mM TCEP. Uncleaved protein was removed by passage over Ni-NTA resin and then diluted 20-fold with Mono-Q binding buffer (50 HEPES pH 8.0). a final purification was performed by loading the diluted sample onto a Mono-Q 5/50 GL column and eluted with a gradient of 50 mM HEPES pH 8.0 and 1 M NaCl. The recombinant kinase domains were buffer exchanged and concentrated by centrifugation to ~ 20 mg/ml in 50 mM HEPES pH 8.0, 100 mM NaCl and 0.1 mM TCEP. The protein was flash frozen in liquid nitrogen and stored at −80 °C.

**Size exclusion chromatography - small-angle X-ray scattering (SEC-SAXS) data collection and analysis**. Small-angle X-ray scattering (SAXS) data were collected at the Stanford synchrotron radiation lightsource BioSAXS beamline BL4-2. Data collection and scattering-derived parameters are described in Supplementary Table 4. The kinase domains of ALK2 and BMPR2 were subjected to investigation, either alone or in complex, at 100 μM in 20 mM HEPES pH 8.0, 100 mM NaCl and 0.1 mM TCEP. Briefly, twin-tandem Superdex 200 3.2/30 GL columns were equilibrated in buffer at room temperature. The sample was injected at a flow-rate of 0.05 ml/min. Scattering images were collected with 1 s exposure every 5 s. Background scatter was determined using the first 100 images and automatically subtracted. Data from five frames were averaged with SasTool[57]. The R$_g$, D$_{max}$, and Porod volume for the peak fractions were calculated using PRIMUS[58] and models were fit with *CRYSOL*[59] in the ATSAS package[60]. Ab initio bead models, of 20 independent runs, were generated using *DAMMIN*[61] and averaged with *DAMAVER*[62]. The crystal structures of ALK2$^{KD}$ (PDB 3MTF), BMPR2$^{KD}$ (PDB 3G2F), and the MD models were superimposed with *SUPCOMB*[63].

**MD simulations**. All simulations were based on X-ray structures of the ALK2$^{KD}$ (PDB code: 3H9R)[26] and BMPR2$^{KD}$ (PDB 3G2F)[35]. Non-protein and non-kinase domain atoms were removed, and the missing loop regions and side-chain atoms were modeled to make the complete protein structures using the software package Maestro (Schrödinger, LLC). The simulations reported here are described in Supplementary Table 5 and the detailed properties of identified protein–protein interfaces are summarized in Supplementary Table 1. Each simulation system was set up by placing it in a cubic simulation box (with periodic boundary conditions) with at least a 10 Å distance from the protein surface to the edge of the simulation box. Na$^+$ and Cl$^-$ ions were added to obtain a neutral total charge for the system and maintain physiological salinity (150 mM). Explicitly represented water molecules were added to fill the simulation box. The systems were each equilibrated on GPU Desmond[64] using a mixed NVT/NPT schedule. MD simulations were performed on the special-purpose machine Anton[65] in the NPT ensemble with $T = 310$ K and $P = 1$ bar using a variant of the Nosé-Hoover and the Martyna-Tobias-Klein algorithms[66–68]. The simulation time step was 2 fs; the r-RESPA integration method was used, with long-range electrostatics evaluated every three time steps. A 9 Å cutoff was applied for the van der Waals calculations. The pairwise summation of electrostatic forces was cut off at 13.7 Å, and long-range electrostatics were computed in $k$-space using a grid-based method (using Gaussian spreading to the grid)[69]. The simulation trajectories were visualized and analyzed using Visual Molecular Dynamics (VMD) software[70], and the images of protein structures were made using the PyMOL Molecular Graphics System (Schrödinger, LLC). a99SB*-ILDN with TIP3P water has been shown to accurately reproduce the structure of protein complexes, however, this force field has the tendency to overstabilize compact conformations[71,72]. We have also found that both ALK2$^{KD}$ and BMPR2$^{KD}$ have large patches of charged surfaces (Fig. 2c). For these reasons, the spontaneous binding simulations of ALK2$^{KD}$ and BMPR2$^{KD}$ used the DES-Amber force field with TIP4P-D water, which when used in combination more accurately represent nonbonded interactions at the protein–protein interfaces[71,73]. These simulations were all initiated from one arbitrary spatial arrangement of ALK2$^{KD}$ and BMPR2$^{KD}$ with different random atomic velocities. This is a standard practice for simulations of spontaneous biomolecular binding[24,25,74], in which the biomolecules quickly diverge from their initial arrangement on the nanosecond timescale (Supplementary Fig. 2B illustrates this for this study). The remainder of the systems in the study used the a99SB*-ILDN force field[75] (which builds on other modifications[76,77] to Amber99[78]) with TIP3P water[79]. For ions, the parameters of Beglov and Roux were used[80]. After the generation of the N:N complex model, the GS domain residues were built in the substrate-binding pocket of BMPR2$^{KD}$ by homology modeling based on crystal structures of PKA (Protein Kinase A, PDB code: 1ATP)[81]. All models underwent a final restrained energy minimization using Maestro.

**H/D exchange mass spectrometry**. H/D exchange time courses were performed in triplicate with manual $D_2O$ dilution, quench, and pepsinization[82]. Deuterium exchange was initiated by diluting protein (250 pmol per time point) into buffered $D_2O$ (50 mM HEPES pD 7.5, 50 mM KCl, 5 mM DTT) to a final deuterium content of 96% and 6.3 µM kinase at 23 °C. At various time points (15, 180, 900 s) exchange was manually quenched by the addition of 5% trifluoroacetic acid (TFA) to the final pH 2.5 (5.1 µL), followed by immediate freezing in liquid $N_2$. Technical replicates were performed by initiating three independent exchange reactions. Time point samples were quickly thawed and digested (3 min, 4 °C) into uniquely identifiable peptides with agarose-immobilized pepsin resin (Thermo Fisher Scientific) in aqueous 0.025% TFA, 6 M urea pH 2.5. Pepsin resin was removed by brief centrifugation (3 s, $10,000 \times g$), and samples were immediately refrozen in liquid $N_2$. Samples were stored at −80 °C until analysis. Pepsin digestion products were identified by LC-MS/MS using undeuterated samples.

For deuterium uptake LC-MS analysis, exchange samples were thawed and resolved over a C8 reversed-phase analytical column (Pinnacle DB 5 µm particle, 30 mm × 1 mm, Restek) via an Agilent 1260 HPLC preequilibrated with 0.1% formic acid/10% acetonitrile. Analytical column, inlet tubing, and injector were maintained at 0 °C–4 °C. Peptides were eluted using a linear gradient from 10–25% acetonitrile over 2.5 min followed by 5.5 min ramp to 55% acetonitrile. Peptides were ionized with a HESI-II electrospray ionization source (Thermo) and analyzed on a Q-Exactive Orbitrap (Thermo) mass spectrometer. Mass spectra were collected in positive-ion mode ($m/z$ range of 300–1800) at a mass resolution setting of 70,000.

For peptic peptide identification, MS/MS data were processed using Proteome Discoverer (Thermo) with Mascot and SEQUEST for peptide identification using 5 ppm mass error tolerance for precursor ions, 0.05 Da tolerance for fragment masses, and no enzyme specified. HDX-MS data were processed and analyzed using HDX Workbench Ver. 2.9.8[83]. Each peptide, time point, replicate, and observed charge state was manually curated. Multiple charge states from the same peptide were analyzed independently. Final data summarie include one representative charge state for each unique peptide. Back-exchange was estimated from peptides encompassing the highly exchange-prone N-termini. Relative deuterium uptake was calculated using HDX Workbench and reported without back-exchange correction. Statistical significance was determined using a two-tailed unpaired T-test for each time point. Deuterium exchange differences are reported for representative time points sampling the actively exchanging regime (approximately the middle 25–75%) of the timecourse. Peptide coverage and deuterium uptake perturbations were mapped onto structural models using PyMol. HDX-MS summary statistics are reported in Supplementary Table 6 in accordance with community-based guidelines[84]. Uptake plots for all HDX experiments are reported in Supplementary Data 3.

**Crystallization and data collection**. The kinase domain of BMPR2 containing the D485G mutation (BMPR2[KD]-D485G) was crystallized with ADP using conditions previously described for wild-type BMPR2 kinase[35] and cryo-protected with mother liquor and 10% ethylene glycol. The kinase domain of ALK2 harboring the single K493A (ALK2[KD]-K493A) was incubated with 10 mM AMPPNP. ALK2[KD]-KK492/3AA was incubated with 10 mM AMPPNP or 1 mM LDN-193189. The complexes were crystalized at 9 mg/ml by sitting drop vapor diffusion at 293 K with 100 nl of protein solution to 100 nl of precipitant solution. Crystallization was successful in 0.05 M PIPES pH 7, 10% w/v PEG 4000 and 10 mM DTT. The crystals were cryo-cooled in liquid nitrogen after supplementing mother liquor with 20% ethylene glycol. Diffraction data were collected at ALS beamline 8.3.1. The data were integrated in XDS[85], scaled, and merged in AIMLESS[86], part of the CCP4 suite version 7.0[87]. Molecular replacement was performed in PHASER version 2-7-17[88] using the BMPR2 kinase domain (PDB 3G2F) or the ALK2 kinase domain (PDB 3MTF). The structures were refined by iterative rounds of refinement in Phenix.refine version 1.13-2998[89] and manual model building with COOT version 08-8-8[90].

**Kinase assay**. Kinase activity of the recombinant protein was measured using a continuous enzyme-coupled reaction system performed at 30 °C[91], with minor modifications. Briefly, the reaction buffer contained 20 mM Tris pH 7.5, 100 mM NaCl, 10 mM $MgCl_2$, 1 mM PEP (Sigma Aldrich), 56 U/ml PK/LDH (Sigma Aldrich), 0.3 mg/ml NADH (Sigma Aldrich) and 1 mM ATP. The reaction was initiated by the addition of recombinant kinase to the final concentration, as indicated in the figure legends. Kinase phosphorylation was followed by recording the enzyme-coupled oxidation of NADH to $NAD^+$, measured at 340 nm, on a VersaMax microplate reader (Molecular Devices), the data were plotted in Prism 7 (GraphPad).

**Cell culture and transfection**. HEK293 cells were purchased from American Type Culture Correction and maintained in Dulbecco's Eagle media (DMEM) containing 10% fetal bovine serum (FBS). HEK293 clones in which pFAm.BRE.Luc. Neo construct[92] was stably integrated (HEK-BRR-luc cells) were maintained in DMEM containing 10% FBS with 500 µg/ml of G418. Cells were cultured at 37 °C in the presence of 5% $CO_2$. Plasmid transfections were performed using Lipofectamine 2000 (Life Technologies) or Lipofectamine 3000 (Life Technologies).

**Immunoblot analysis and antibodies**. Cells were lysed in the RIPA lysis buffer containing 150 mM NaCl, 1% NP-40, 0.5% sodium deoxycholate, 0.1% SDS, 50 mM Tris pH 8.0, and incubated for 15 min at 4 °C. Lysates were cleared by centrifugation, separated by SDS-PAGE, transferred to nitrocellulose membrane (Millipore), and immunoblotted with antibodies, followed by visualization by LI-COR imaging system. The antibodies used included anti-BMPR2 carboxyl terminal domain (612292, BD Bioscience), anti-GAPDH (MAB374, Millipore) and horseradish peroxidase (HRP)-conjugated anti-HA antibody (clone 3F10, Roche). SuperSignal West Dura Extended Duration Substrate (Thermo Fisher Scientific) was used as chemiluminescent HRP substrates. The uncropped and unprocessed Western blots are provided in the Source Data file.

**Luciferase assay**. HEK-BRR-luc cells were seeded in 24-well dish at a density of $7 \times 10^3$ cells per well, cultured for 48 h, and transfected with respective plasmids using Lipofectamine 3000. Twenty-four hours after transfection, cell culture media was replaced with 0.2% FBS containing media followed by treatment with or without 300 pM BMP4 (R&D) for 24 h. Cells were rinsed with PBS and subjected to the luciferase assay using the Dual-Luciferase Reporter Assay System (Promega). The thymidine kinase promoter-Renilla luciferase reporter plasmid (pRL-TK, Promega) was co-transfected and used for normalization of the transfection efficiency.

**Statistical analysis**. Statistical Analysis was performed using the Prism 7.0 GraphPad package. Statistical tests and significance are denoted in the figures and figure legends.

**Reporting summary**. Further information on research design is available in the Nature Research Reporting Summary linked to this article.

## Data availability

Atomic coordinates have been deposited in the Protein Data Bank (PDB) under accession codes 6UNP [https://doi.org/10.2210/pdb6UNP/pdb] (BMPR2KD-D485G), 6UNQ [https://doi.org/10.2210/pdb6UNQ/pdb] (ALK2KD-K493A with AMPPNP), 6UNR [https://doi.org/10.2210/pdb6UNR/pdb] (ALK2KD-K492A/K493A with AMPPNP) and 6UNS [https://doi.org/10.2210/pdb6UNS/pdb] (ALK2KD-K492A/K493A with LDN-193189). The MD trajectories for data described in Figs. 2 and 7a–c and Supplementary Figs. 2, 10, and 12 are available for non-commercial use through contacting trajectories@deshawresearch.com. Atomic coordinates for the reported models are provided in Supplementary Data 1 and 2. All HDX-MS uptake plots for data presented in Fig. 3 and Supplementary Figs. 4, 6, 7, and 8 are included in Supplementary Data 3. HDX-MS data files and source data for Supplementary Figs. 4, 6, 7, and 8 are available in the ProteomeXchange Consortium via the PRIDE[93] partner repository with the dataset identifier PXD022944. The molecular dynamics (MD) simulations were performed using the Anton 2 supercomputer. The simulation code we used is specialized to Anton 2, but codes for performing MD simulation are widely available. Source data are provided with this paper.

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

## Acknowledgements

H.S.L. and E.S.U. were supported by Grant 1715411 from the National Science Foundation, Division of Molecular and Cellular Biosciences. We thank Joel Nott of the Protein Facility of the Iowa State University Office of Biotechnology for technical support. We thank Joan Massague of MSKCC for the HA-ALK2 construct. This study was funded by NIH (HL132058) to A.H. We also thank the staff at the Advanced Light Source Beam Line 8.3.1, Lawrence Berkeley National Laboratory and BioSAXS Beam Line 4-2, Stanford Synchrotron Radiation Lightsource.

## Author contributions

C.A., A.H., and N.J. designed the project. C.A., P.A., A.H., E.S.U., and N.J. wrote the manuscript. C.A. and J.E.K. designed, expressed, and purified the constructs. C.A. performed and analyzed the SEC-SAXS data, crystal structures, and kinase assays. P.A., Y.S., and D.E.S. designed, performed, and analyzed the MD simulations. R.K. and P.G. performed and analyzed the immunoblots, qRT-PCR, and luciferase assays. H.S.L. and E.U. performed and analyzed the HDX-MS data. All authors discussed the results and commented on the manuscript.

## Competing interests

The authors declare no competing interests.
