## [Peer Review File · Nature Communications]

Reviewers' Comments:

Reviewer #1:

Remarks to the Author:

Agnew, Ayaz et al. characterize the heterodimeric complex of the kinase domains of two very interesting receptor kinases, ALK2 and BMPR2, and relate these complexes to the previously-unknown activation mechanism that is presumed to arise from tetramerization. By combining the component domain structures with MD and HDX-MS, the authors have been able to provide a clear explanation for how ALK2 and BMPR2 can reside in a dormant state until activated by ligands at the receptor ectodomain. It is surprising that their SAXS data were not used for further validation; I suggest the authors extend their analysis to explore this possibility so as to provide further support for the modeled C-lobe to C-lobe pose of heterodimerization. Nonetheless, by testing their dimerization model by mutational analyses with recombinant proteins and in cells, and extending the cellular analyses to test the tetrameric assembly proposed, the authors provide a compelling case for these assemblies and how the catalytic inhibition imparted by the GS domain might be relieved. Additionally, the authors are able to relate the model to patient variations, which provide important support for the proposed assembly and important insights into the molecular basis of human disease. I suggest some minor analyses and discussion points below that may improve the communication of the data.

Major points:

1. The abstract does not capture that SAXS was also performed. Further, as mentioned above, it is not clear why the authors did not compare their HDX/MD derived model with the SAXS data collected on the protein complex. A CRYSOLOG comparison of coordinates is warranted.
2. The first paragraph of the introduction requires referencing. Page 5 paragraph 2 – primary citations would be more appropriate than solely including a review on the structures. Although perhaps the number of references overall might have been a factor.
3. End intro, bottom page 5. Not clear why GS domain positioned away from BMPR2 active site would suppress activation. Some elaboration is required. My interpretation is that this prevents their phosphorylation, but in the context of this description, the reader may not be aware that the GS domains are only suppressive in their dephosphorylated forms. I suggest mentioning their positioning prevents autophosphorylation and relief of suppression.
4. Figure 1. The authors do not present scattering data. Considering the sensitivity of SAXS to interference from polydispersity, it is essential that the authors present their scatter and Guinier plots. While not overly rigorous, at a minimum I suggest including a table to summarise the data as per DA Jacques Acta Cryst D 2012. There is a move towards reporting standards that are substantially more detailed, but I feel Jacques 2012 is sufficient.
5. Figure 1 and elsewhere. It would be helpful if the authors could mark the elution positions of MW standards above SEC profiles
6. Figure 5c requires inclusion of molecular weight marker positions. It would be helpful if the antibodies used for detection were noted in the figure next to the respective panels
7. Mutant receptors introduced into cells: the authors only consider these in the context of the heterodimer, although it is clear there is greater complexity. I feel it is important to acknowledge there may be an impact on the tetramerization in the discussion.
8. While acknowledging the extensive data (20 overall figures), the text in the manuscript could be presented in a more concise manner. The Discussion in particular could be trimmed. Perhaps this would allow space for the authors to consider comparing the kinase-kinase interaction mode to RTKs beyond the ALK family, and could consider the role of Juxtamembrane sequences in modulating kinase

activities (e.g. Eph receptors). Such comparisons might allow some broader messages about kinase activation mechanisms to be communicated.

Minor:

9. Inconsistent use of HDX-MS vs HDX/MS

10. Methods. QuikChange rather than QuickChange. I never really understood why, except maybe the naming rights belong to Bill Murray?

11. End page 26 – 2 Protease inhibitor tablets in what volume? Supplier of Ni resin? Same queries for BMPR2 purifications on page 27.

12. For SAXS data: what was used to estimate background scatter? And how was background subtracted? What temperature was used for data collection?

13. Were HDX timepoints enforced manually or robotically?

14. Kinase assay: it would be helpful to include details of detection.

15. Code availability. Should this be "have" rather than "gave"?

Reviewer #2:

Remarks to the Author:

Comments to the authors

The authors have presented a very engaging story describing the oligomerization of type I/type II kinases. They have used a combination of structural analysis techniques that are quickly becoming fast friends that has led to a convincing collection of data. I do wonder if this manuscript is more suitable to a journal with a more structure/function/mechanistic focus such as Nature Structure Molecular Biology. Regardless, I would suggest some modifications that would be triggered by my comments described below.

General comments regarding the HDX presented here:

The authors have not completely conformed to the modern standard adopted by the HDXMS community for data reporting. The following will address this problem: Upload raw MS files, peptide lists, and summary processed data to the PRIDE archive, or similar, as is now customary for submission to Nature journals. It would also be helpful to describe the analysis and processing in greater detail, which can be done in the summary of the PRIDE submission (see example PRIDE entries PXD013051 or PXD011060).

It would be helpful to have a peptic peptide map with amino acid numbering so that the deuterium incorporation plots can be compared to what is presented in Figure 3, the Supplementary Figures 3, 5, and 6 as well as regions called out in text. It would also be helpful to see the HDX differences in an un-condensed representation in the supplementary figures.

Specific comments

Page 9: "...these regions exhibited ~28% decrease in the extent of deuterium exchange rate..." I don't believe that the authors fit the deuterium incorporation curves to a multi-exponential equation and therefore haven't calculated an exchange rate and therefore a difference in exchange rate.

Additionally, I would argue that considering the data were collected with only 3 time points any extraction of exchange rates would not be possible. It is clear from the deuterium incorporation plots that there are differences in deuterium incorporation for these regions that the authors have highlighted on the models. I would therefore suggest picking a single time point and plotting the raw differences in percent exchange (or otherwise) on the models and then showing the balance of the data in the supplementary figures. This comment also will extend to the remainder of the HDX data presented in the manuscript.

Page 10: "However, no significant HDX/MS differences were measured on the ALK2KD side of the N dimer interface, arguing against formation of the N dimer in solution." I would counter argue that

simply because there aren't any changes to the HDX/MS profile of the ALK2 kinase domain doesn't mean that there isn't an interaction. There are plenty of examples in the literature that show no changes in HDX profiles for one binding partner in a complex while the other shows dramatic changes. There certainly could be an interaction via the N-lobes of the kinases in which there is/are no alteration/s to the backbone hydrogen bonding network in the N-lobe of the ALK2 kinase domain. Was there any evidence of multiple populations in the raw mass spectral data that could be seen for any of the peptides, especially in the N-lobe?

Although not explicitly obvious from the separate HDX summary data tables 4a-c is it true that there were three independent HDX experiments? Would it be possible to compare/show with deuterium incorporation plots and in Supplementary Figure 6, GS-ALK2KD vs BMPR2KD:GS-ALK2KD and BMPR2KD vs BMPR2KD:GS-ALK2KD. For this reviewer at least, this would eliminate the mental gymnastics needed to compare the different constructs of ALK2? This exercise is done for Supplementary Figure 10 and then it would remain consistent throughout.

The augmented changes in HDX for the N-lobe of BMPR2 kinase domain in the heterodimer complex once the GS-domain is included in the ALK2 construct are very interesting. How do changes in the N-lobe of the ALK2 kinase domain (GS domain binding) translate through the bound C-lobes of the BMPR2KD:ALK2KD complex to the N-lobe of the BMPR2 kinase domain? How do the authors rationalize the differences seen in the N-lobe of BMPR2 in the BMPR2KD:ALK2KD HDX results in context of HDX data that have been published for other kinase domains?

The HDX changes plotted on the C2 tetramer model are very interesting. Does this help to clarify the discrepancies seen when the data are plotted on one model of the dimer vs the other (p. 17 "Although the C2 dimer is less compatible with the HDX/MS data overall than the C1 dimer...")? It appears that plotting these data on the tetrameric model make greater sense of the HDX changes in the N-lobe of the kinase domain. Would you think then that the HDX experiment is revealing a transient tetramer that that isn't stable enough to see in other solution analyses? Combined with the comment about the argument made on page 10 (above) this could certainly be a possibility. Along these lines, as it is not clear from the HDX/MS methods section, at what protein concentrations were the HDX labeling experiments performed?

Reviewer #3:

Remarks to the Author:

The manuscript "Structural basis for ALK2/BMPR2 receptor complex signaling through kinase domain oligomerization" reports an investigation of hetero-dimerization of the Ser/Thr kinase domains of the transmembrane receptors type I ALK2 and type II BMPR2. This is an interesting story in which MD simulations were used to identify a potential protein-protein interface, and HDX MS, SEC and cell studies were used to validate the interface and its biological relevance. The study addresses an important question of what how intermolecular association of intracellular domains, triggered by extracellular ligand binding, transduce signals. The experimental work provides solid evidence for the proposed model. The study is thorough and the presentation thoughtful. Nonetheless, not everything is clear and the authors should respond to the comments below

- The study utilizes the kinase domains specified in fig 1b. The BMPR2 construct excludes ~500-residue region C-terminal to the KD (residues 510 to 1038). What is known structurally about this large Cter region? Is it reasonable to assume that heterodimerization involving the C-lobe of the KD is unaffected by this region in the full length protein?

- It appears that the 40 MD simulations differed only by initial velocities. An interesting, and more to-the-point parameter to vary for initial conditions would be the relative orientation of the two KDs. The resulting dimer interactions must depend on the initial relative orientation given 20 μ s sampling time

period. How was this initial orientation decided and were alternative orientations tried?

- Little information is provided on the behavior of dimer interactions over the 40 trajectories. Fig 2a visualizes the final structures. From the broad distribution of orientations in the middle circled-2 panel, the dimeric association looks stochastically distributed. Why consider only the final frames of the trajectories? It would seem that many encounters occur in a 20- μ s trajectory and the final frames are not necessarily to most relevant complexes. Second, this distribution appears to have BMPR2 oriented around ALK2 roughly in the plane of the paper, not spherically distributed over an entire 360 degrees. Is this restriction due to the geometry of the simulation box?

- So how were these "interesting" (page 7) N dimer and C1 dimer identified? It would be useful for the authors to show an analysis of the 40 trajectories to indicate the probability of a dimeric orientation and the lifetime/time period of the encounters. What is the time-dependent behavior over the course of the MD trajectories, and why examine only the final frames (fig 2a)? Supplementary figure 2 may report something along these lines but there was insufficient description of this figure to understand what is being plotted. (no definition of RMSD and no indication of what trajectory). Further, the other 16 trajectories should be plotted.

- What is Fig 2b? The caption needs clarification. A histogram plot is shown but there are no "final C1-dimer and N-dimer models." RMSD needs to be defined; what atoms, what reference structure, and what atoms are superpositioned before calculating the difference? And the trajectory period for accumulating the histogram values needs defining. Second, stability seems to be defined as a narrow RMSD distribution, as opposed to stability in a thermodynamic sense. Clearly, thermodynamic stability is not the case here. The word "stable" needs to be qualified. Further, the meaning of stability seems to differ within the ms; stable meaning lifetime for MD and stable meaning exchange rates for HDX-MS. More explicit wording would be helpful.

- Several references to Fig 2b throughout the paper seem to refer to some other figure. e.g. on page 8: "once the kinases adopted this pose, it was largely stable for the remainder of the simulation (Fig. 2b)" Fig 2b has no time dependence -- ?? Similar disconnects appear on p. 9, 11, etc.

- Can the authors comment on their MD approach to identify a biologically relevant dimer, supported with strong experimental data, in contrast to association of almost any pair of proteins with a certain amount of surface area of opposite electrostatic charge under a similar setup for MD (very high concentrations).

- What are the structures deposited in each PDB entry 6UNS, 6UNQ, 6UNR and 6UNP?

- Supplementary Table 2 lists a number of simulations that we not referenced (as far as I could tell) in the manuscript. In particular, the simulations with the motivation of assessing "stability" of a dimer. Is the text referencing this table inadvertently missing from the main manuscript?

REVIEWER COMMENTS

Reviewer #1 (Remarks to the Author):

Agnew, Ayaz et al. characterize the heterodimeric complex of the kinase domains of two very interesting receptor kinases, ALK2 and BMPR2, and relate these complexes to the previously-unknown activation mechanism that is presumed to arise from tetramerization. By combining the component domain structures with MD and HDX-MS, the authors have been able to provide a clear explanation for how ALK2 and BMPR2 can reside in a dormant state until activated by ligands at the receptor ectodomain. It is surprising that their SAXS data were not used for further validation; I suggest the authors extend their analysis to explore this possibility so as to provide further support for the modeled C-lobe to C-lobe pose of heterodimerization. Nonetheless, by testing their dimerization model by mutational analyses with recombinant proteins and in cells, and extending the cellular analyses to test the tetrameric assembly proposed, the authors provide a compelling case for these assemblies and how the catalytic inhibition imparted by the GS domain might be relieved. Additionally, the authors are able to relate the model to patient variations, which provide important support for the proposed assembly and important insights into the molecular basis of human disease. I suggest some minor analyses and discussion points below that may improve the communication of the data.

We thank the reviewer for consideration of our manuscript and have addressed the points they have raised below.

Major points:

1. The abstract does not capture that SAXS was also performed. Further, as mentioned above, it is not clear why the authors did not compare their HDX/MD derived model with the SAXS data collected on the protein complex. A CRYSOLOG comparison of coordinates is warranted.

We have incorporated SAXS into the abstract and we now include a comparison of the models to the experimental SAXS data.

2. The first paragraph of the introduction requires referencing. Page 5 paragraph 2 – primary citations would be more appropriate than solely including a review on the structures. Although perhaps the number of references overall might have been a factor.

We agree with the reviewer that primary citations would be more appropriate, however the extensive body of work in this area, makes it prohibitive to include all the citations. Since we already have 95 references associated with this manuscript, we hope that the reviewer agrees that a comprehensive review article that properly acknowledges all these contributions is the most appropriate reference covering this complex and extensive topic.

3. End intro, bottom page 5. Not clear why GS domain positioned away from BMPR2 active site would suppress activation. Some elaboration is required. My interpretation is that this prevents their phosphorylation, but in the context of this description, the reader may not be aware that the GS domains are only suppressive in their dephosphorylated forms. I suggest mentioning their positioning prevents autophosphorylation and relief of suppression.

To increase clarity of the section describing role of the C-lobe heterodimer in preventing GS domain phosphorylation we have incorporated the reviewer's suggestion in the revised manuscript.

4. Figure 1. The authors do not present scattering data. Considering the sensitivity of SAXS to interference from polydispersity, it is essential that the authors present their scatter and Guinier plots. While not overly rigorous, at a minimum I suggest including a table to summarise the data as per DA Jacques Acta Cryst D 2012. There is a move towards reporting standards that are substantially more detailed, but I feel Jacques 2012 is sufficient.

We thank the reviewer for their suggestion and in the revised manuscript we now include a supplemental table detailing the SAXS collection and scattering parameters. We have also included a more detailed analysis of the scattering particles that demonstrates that the ALK2/BMP2 oligomeric state is consistent with a dimeric state in solution. In Figure 2 we have added a panel showing the scattering profile of the ALK2/BMP2 complex including the *CRY SOL* fits of the N dimer and C1 dimer. Either side are the averaged *ab initio* models from *DAM AVER* with the respective models superimposed by *SUP COMB*.

To give the reader a complete view of the scattering data we have included additional plots suggested by the reviewer in reference to the published study (*DA Jacques Acta Cryst D 2012*). Supplementary figure 1 shows the *Log X-Log Y* plot to extenuate the low-angle data of the complex. We have included the Guinier plot, which together, shows the complex is homogenous and does not show features that could be attributed to aggregation. In addition to the added panels in Figure 2, we also provide a plot of the scattering curve with fits for each of the models described in this paper and show their χ^2 value. These plots demonstrate that dimeric models fit the scattering data most accurately while the monomeric kinases, or the tetramer model, produce poor fits to the data. We also fit the *DAM AVER* bead model to the molecular dynamics models, which provides another support of this conclusion. As the reviewer noted, SAXS is very sensitive to particle aggregation and if the ALK2 and BMP2 kinases were forming a higher order tetrameric complex, then this would have been detected in the scattering analysis.

5. Figure 1 and elsewhere. It would be helpful if the authors could mark the elution positions of MW standards above SEC profiles

To aid with understanding we have added molecular weight standards to Figure 1. We have not included these in Figures 4 and 6 as they contain a reference peak corresponding to the complex, which is shown on each graph.

6. Figure 5c requires inclusion of molecular weight marker positions. It would be helpful if the antibodies used for detection were noted in the figure next to the respective panels

The molecular weight markers have been marked in Fig. 5c, and also in Fig 7d.

7. Mutant receptors introduced into cells: the authors only consider these in the context of the heterodimer, although it is clear there is greater complexity. I feel it is important to acknowledge there may be an impact on the tetramerization in the discussion.

We thank the reviewer for highlighting this critical point and we have included additional language in the discussion to emphasize the impact of these mutations on receptor tetramerization and signaling.

8. While acknowledging the extensive data (20 overall figures), the text in the manuscript could be presented in a more concise manner. The Discussion in particular could be trimmed. Perhaps this would allow space for the authors to consider comparing the kinase-kinase interaction mode to RTKs beyond the ALK family, and could consider the role of Juxtamembrane sequences in modulating kinase activities (e.g. Eph receptors). Such comparisons might allow some broader messages about kinase activation

mechanisms to be communicated.

In line with the reviewer's comment we have trimmed the Results and Discussion sections wherever possible, and the latter in part by extracting the discussion of disease mutations in a separate section of Results. We have also included a short discussion comparing kinase activation mechanisms between RTKs and RSTKs.

Minor:

9. Inconsistent use of HDX-MS vs HDX/MS

We thank the reviewer for pointing this out, and have updated to a consistent HDX-MS.

10. Methods. QuikChange rather than QuickChange. I never really understood why, except maybe the naming rights belong to Bill Murray?

Amended in the text.

11. End page 26 – 2 Protease inhibitor tablets in what volume? Supplier of Ni resin? Same queries for BMPR2 purifications on page 27.

We have included the requested information in the methods section.

12. For SAXS data: what was used to estimate background scatter? And how was background subtracted? What temperature was used for data collection?

We have added the relevant information to the methods section.

13. Were HDX timepoints enforced manually or robotically?

HDX timepoints were performed manually. We have updated the Methods to clarify this.

14. Kinase assay: it would be helpful to include details of detection.

The details of kinase assay detection have been added to the methods.

15. Code availability. Should this be "have" rather than "gave"?

This statement was edited in the revised manuscript.

Reviewer #2 (Remarks to the Author):

Comments to the authors

The authors have presented a very engaging story describing the oligomerization of type I/type II kinases. They have used a combination of structural analysis techniques that are quickly becoming fast friends that has led to a convincing collection of data. I do wonder if this manuscript is more suitable to a journal with a more structure/function/mechanistic focus such as Nature Structure Molecular Biology. Regardless, I would suggest some modifications that would be triggered by my comments described below.

General comments regarding the HDX presented here:

The authors have not completely conformed to the modern standard adopted by the HDXMS community for data reporting. The following will address this problem: Upload raw MS files, peptide lists, and summary processed data to the PRIDE archive, or similar, as is now customary for submission to Nature journals. It would also be helpful to describe the analysis and processing in greater detail, which can be done in the summary of the PRIDE submission (see example PRIDE entries PXD013051 or PXD011060).

We thank the reviewer for the PRIDE submission recommendations and examples. We have uploaded raw MS files, peptide lists, and HDX Workbench processed data to the PRIDE archive under the dataset identifier PXD022944. While under review, the data can be accessed using the following credentials:

Username: “reviewer_pxd022944@ebi.ac.uk”, Password: “wyRkgkpz”. As the reviewer recommended, we also took advantage of this opportunity to expand the analysis methods in the PRIDE Sample Processing and Data Processing Protocols sections.

It would be helpful to have a peptic peptide map with amino acid numbering so that the deuterium incorporation plots can be compared to what is presented in Figure 3, the Supplementary Figures 3, 5, and 6 as well as regions called out in text. It would also be helpful to see the HDX differences in an uncondensed representation in the supplementary figures.

We have updated Supplementary Figures 4, 6, 7 and 8 to include amino acid numbering and the full complement of peptic peptides to correspond to the deuterium incorporation plots. Figure legends have also been updated to reflect the expanded data.

Specific comments

Page 9: “...these regions exhibited ~28% decrease in the extent of deuterium exchange rate...” I don’t believe that the authors fit the deuterium incorporation curves to a multi-exponential equation and therefore haven’t calculated an exchange rate and therefore a difference in exchange rate. Additionally, I would argue that considering the data were collected with only 3 time points any extraction of exchange rates would not be possible. It is clear from the deuterium incorporation plots that there are differences in deuterium incorporation for these regions that the authors have highlighted on the models. I would therefore suggest picking a single time point and plotting the raw differences in percent exchange (or otherwise) on the models and then showing the balance of the data in the supplementary figures. This comment also will extend to the remainder of the HDX data presented in the manuscript.

We agree that this sentence generates confusion about the data analysis. The reviewer is entirely correct that the incorporation plots were not fit to a kinetic model. As such, we are not really comparing rate changes. Rather, as the reviewer recommends, we were relying on representative time points (sampling, where possible, the time points closest to the midpoint of the exchange progression, i.e., ~25 – 75% exchanged). We agree with the reviewer’s recommendations for plotting raw differences for a representative point. Indeed, this is exactly how the figure color-coding and mapping was done. We have clarified this point in the updated data processing methods. In addition, we have revised the ‘rate decrease’ wording to more accurately reflect the data analysis as we performed it-- and as recommended by the reviewer.

Page 10: “However, no significant HDX/MS differences were measured on the ALK2KD side of the N dimer interface, arguing against formation of the N dimer in solution.” I would counter argue that simply because there aren’t any changes to the HDX/MS profile of the ALK2 kinase domain doesn’t mean that there isn’t an interaction. There are plenty of examples in the literature that show no changes in HDX profiles for one binding partner in a complex while the other shows dramatic changes. There certainly could be an interaction via the N-lobes of the kinases in which there is/are no alteration/s to the backbone hydrogen bonding network in the N-lobe of the ALK2 kinase domain. Was there any evidence of multiple populations in the raw mass spectral data that could be seen for any of the peptides, especially in the N-lobe?

We thank the reviewer for pointing out this important note of caution in interpreting the lack of exchange rate perturbations. We agree that there are several possible mechanisms by which amide proton exchange could remain undetectable at a binding interface. We have modified the rationale in the main text to focus on strong exchange perturbations observed at the C-lobe interface. Furthermore, we have added a note acknowledging that a lack of exchange perturbation alone does not rule out an interface. Regarding the interesting question about multiple populations evident in the raw data, we agree that this is an intriguing idea. We have re-examined the spectra, but we don’t see much evidence for multiple populations in the isotope envelopes.

Although not explicitly obvious from the separate HDX summary data tables 4a-c is it true that there were three independent HDX experiments? Would it be possible to compare/show with deuterium incorporation plots and in Supplementary Figure 6, GS-ALK2KD vs BMPR2KD:GS-ALK2KD and BMPR2KD vs BMPR2KD:GS-ALK2KD. For this reviewer at least, this would eliminate the mental gymnastics needed to compare the different constructs of ALK2? This exercise is done for Supplementary Figure 10 and then it would remain consistent throughout.

That is correct, there were three independent HDX-MS experiments. We appreciate the recommendation for clarifying the scope of the HDX-MS experiments by adding the comparisons suggested by the reviewer. We have added the HDX-MS comparisons of GS-ALK2KD vs. the GS-ALK2KD:BMPR2 complex and the BMPR2 vs. the GS-ALK2KD:BMPR2 complex in the Supplementary Fig. 8. This is accompanied by an additional HDX Summary Table and submission of the raw data, etc. to the PRIDE depository.

The augmented changes in HDX for the N-lobe of BMPR2 kinase domain in the heterodimer complex once the GS-domain is included in the ALK2 construct are very interesting. How do changes in the N-lobe of the ALK2 kinase domain (GS domain binding) translate through the bound C-lobes of the BMPR2KD:ALK2KD complex to the N-lobe of the BMPR2 kinase domain? How do the authors rationalize the differences seen in the N-lobe of BMPR2 in the BMPR2KD:ALK2KD HDX results in context of HDX data that have been published for other kinase domains?

We agree with the reviewer that this is an intriguing observation, and, in our opinion, the most plausible interpretation of these data is in context of the tetramer model, as insightfully suggested in the following reviewer's comment. Our interpretation is therefore elaborated more in the answer to the comment below.

The HDX changes plotted on the C2 tetramer model are very interesting. Does this help to clarify the discrepancies seen when the data are plotted on one model of the dimer vs the other (p. 17 "Although the C2 dimer is less compatible with the HDX/MS data overall than the C1 dimer...")? It appears that plotting these data on the tetrameric model make greater sense of the HDX changes in the N-lobe of the kinase domain. Would you think then that the HDX experiment is revealing a transient tetramer that that isn't stable enough to see in other solution analyses? Combined with the comment about the argument made on page 10 (above) this could certainly be a possibility. Along these lines, as it is not clear from the HDX/MS methods section, at what protein concentrations were the HDX labeling experiments performed?

We appreciate reviewer's insightful commentary on the possible link between the HDX-MS data and the tetramer model. The apparent stabilization of the BMPR2 N-lobe in the presence of the GS domain of ALK2 may have arisen due to GS domain bridging interactions that interlock N-lobes of the GS-ALK2 and BMPR2 kinase domains. We favor this model, as it is consistent with the molecular dynamics simulations. We agree that at the protein concentrations during deuterium labeling (6 μ M each), it is entirely possible that the N-lobe stabilization could be a manifestation of a significant population of tetramer in the labeling solution. (We have added protein concentrations to the Methods section to clarify this). Indeed, while the kinase interactions within the tetramer may be transient in solution, their apparent low affinity would be offset in the context of full-length receptors which are embedded in the membrane via their transmembrane domains. We have highlighted this possibility in the context of the reviewer's observations in our Discussion section.

Reviewer #3 (Remarks to the Author):

The manuscript “Structural basis for ALK2/BMP2 receptor complex signaling through kinase domain oligomerization” reports an investigation of hetero-dimerization of the Ser/Thr kinase domains of the transmembrane receptors type I ALK2 and type II BMP2. This is an interesting story in which MD simulations were used to identify a potential protein-protein interface, and HDX MS, SEC and cell studies were used to validate the interface and its biological relevance. The study addresses an important question of what how intermolecular association of intracellular domains, triggered by extracellular ligand binding, transduce signals. The experimental work provides solid evidence for the proposed model. The study is thorough and the presentation thoughtful. Nonetheless, not everything is clear and the authors should respond to the comments below

We thank the reviewer for the positive comments about our manuscript and are providing clarifications that address points of the confusion below.

- The study utilizes the kinase domains specified in fig 1b. The BMP2 construct excludes ~500-residue region C-terminal to the KD (residues 510 to 1038). What is known structurally about this large Cter region? Is it reasonable to assume that heterodimerization involving the C-lobe of the KD is unaffected by this region in the full length protein?

The C-terminal tail of BMP2 is not conserved across the species, and is predicted to be largely unstructured, containing stretches of low-complexity regions thus making it very unlikely to be engaged in protein-protein interactions. The tail has a number of characterized and predicted sites for post-translational modifications, which are considered to serve as docking sites for effector proteins. At present, there is no evidence or even a plausible prediction that the C-terminal role might play a role in receptor oligomerization. Thus, we have excluded it from our constructs, additionally motivated by our observation that in the presence of the C-terminal tail, the yields of BMP2 kinase-containing construct expression and purification were significantly decreased.

All type I and type II BMP and TGF β receptors form active complexes via ligand-induced tetramerization but have negligible C-terminal vestiges. Only BMP2 features the uniquely long C-terminal tail. Thus, by excluding the tail, we have focused on the core kinase interactions that drive assembly of the active receptor complexes. This does not preclude the potential role that the C-terminal tail of BMP2 plays in modulation of these interactions specifically in BMP2-containing complexes. This topic is of big interest to our groups and currently under active investigation.

- It appears that the 40 MD simulations differed only by initial velocities. An interesting, and more to-the-point parameter to vary for initial conditions would be the relative orientation of the two KDs. The resulting dimer interactions must depend on the initial relative orientation given 20 μ s sampling time period. How was this initial orientation decided and were alternative orientations tried?

A single initial pose was indeed used, but in our simulations the two proteins quickly (on the sub-nanosecond timescale) diverged from this initial pose, randomly sampling a multitude of poses and making variation of the initial pose unnecessary. (For this reason, picking a single starting pose is a standard practice for such simulations, and in the revised manuscript we add references related to this point: Shan et al., *JACS* 2011; Shan et al., *NSMB* 2014; Pan et al., *PNAS* 2019.) The rapid sampling of a multitude of diverse relative orientations is reflected in the very large RMSDs with respect to the initial pose in the early phases of the simulations (new Supplementary Fig. 2a,c). In the revised Methods section, we now describe and justify our simulation setup in more detail. We have also added a figure panel (new Supplementary Fig. 2b) that shows the poses of the proteins in our 40 simulations at the 50 ns mark: As can be seen in that figure, the simulations sample a multitude of poses.

- Little information is provided on the behavior of dimer interactions over the 40 trajectories. Fig 2a

visualizes the final structures. From the broad distribution of orientations in the middle circled-2 panel, the dimeric association looks stochastically distributed. Why consider only the final frames of the trajectories? It would seem that many encounters occur in a 20- μ s trajectory and the final frames are not necessarily to most relevant complexes. Second, this distribution appears to have BMPR2 oriented around ALK2 roughly in the plane of the paper, not spherically distributed over an entire 360 degrees. Is this restriction due to the geometry of the simulation box?

We thank the reviewer for this question. Our consideration was not in fact limited to the final poses of the 40 simulations. We considered any protein-protein poses that had remained unchanged in the simulations for more than 5 microseconds. We have revised the text to state this explicitly, and have expanded our discussion of the analyses led us to focus on the N dimer and the C1 dimer.

Regarding Figure 2a, the actual distribution is not confined to a 2D plane, although it may appear to be in the figure (due to the fact that we rendered BMPR2 as partially transparent in order for ALK2 at the center to be visible). We have revised the Figure 2 caption to state this point explicitly.

- So how were these “interesting” (page 7) N dimer and C1 dimer identified? It would be useful for the authors to show an analysis of the 40 trajectories to indicate the probability of a dimeric orientation and the lifetime/time period of the encounters. What is the time-dependent behavior over the course of the MD trajectories, and why examine only the final frames (fig 2a)? Supplementary figure 2 may report something along these lines but there was insufficient description of this figure to understand what is being plotted. (no definition of RMSD and no indication of what trajectory). Further, the other 16 trajectories should be plotted.

In the revised manuscript, we now explicitly describe how the RMSDs were calculated where they are mentioned in the text, and we have added the remaining 36 plots to Supplementary Fig. 2c. As noted in an earlier response, any ALK2-BMPR2 poses that lasted more than 5 microseconds in the 40 simulations were considered “interesting,” and we have added a sentence to the main text to clarify this point. We have also elaborated on the rationale—based on symmetry and other factors—that led us to focus on the N-lobe and C1 dimers. We now also report the interface specifications we used in the analysis of these relatively stable poses in the new Supplementary Table 4.

- What is Fig 2b? The caption needs clarification. A histogram plot is shown but there are no “final C1-dimer and N-dimer models.” RMSD needs to be defined; what atoms, what reference structure, and what atoms are superpositioned before calculating the difference? And the trajectory period for accumulating the histogram values needs defining. Second, stability seems to be defined as a narrow RMSD distribution, as opposed to stability in a thermodynamic sense. Clearly, thermodynamic stability is not the case here. The word “stable” needs to be qualified. Further, the meaning of stability seems to differ within the ms; stable meaning lifetime for MD and stable meaning exchange rates for HDX-MS. More explicit wording would be helpful.

We thank the reviewer for this comment, and we have revised the Fig. 2b caption for clarity. The histograms are based on simulations initiated from the C1 or N dimers generated from the 40 free-association simulations. RMSDs with respect to the initial C1 or N dimer structure were calculated and represented as histograms, which are used as indicators of the conformational stability of the dimers. We have now added a Supplementary Fig. 12, in which the RMSDs as functions of time are plotted. When used without a qualifier, we typically use the word “stable” to describe conformational stability of a protein or protein-protein complex in simulation, which is typically assessed using RMSD. We have checked our usage of this word in the text for consistency, and qualifiers have been added when necessary (e.g., “thermodynamic stability” is used to refer to the strength of the dimerization).

- Several references to Fig 2b throughout the paper seem to refer to some other figure. e.g. on page 8: “once the kinases adopted this pose, it was largely stable for the remainder of the simulation (Fig. 2b)” Fig 2b has no time dependence -- ?? Similar disconnects appear on p. 9, 11, etc.

We thank the reviewer for calling our attention to these errors, which we have remedied in the revised manuscript. We also added the raw RMSD data that is the basis for Fig 2b in a new Supplementary Fig. 12.

- Can the authors comment on their MD approach to identify a biologically relevant dimer, supported with strong experimental data, in contrast to association of almost any pair of proteins with a certain amount of surface area of opposite electrostatic charge under a similar setup for MD (very high concentrations).

A main advantage of an MD-based approach is that in such simulations proteins are not rigid bodies. They remain flexible, as they are in solvent, and can “breathe”. This allows the exploration of protein-protein poses that are not accessible using approaches that treat proteins as rigid objects. The long timescale of the simulations is also important, because it takes time for proteins to escape from their initial conformations. We do not use our MD approach on its own, however: MD used alone can generate biologically irrelevant dimers, and there are no reliable and systematic ways to filter them out. That is why, to be successful, such simulation studies need to begin with biologically sound premises and use experimental work for validation. Our experience so far has indicated that this approach can generate biologically relevant dimers that can be identified with help from experimental data (Shan et al., *NSMB* 2014).

- What are the structures deposited in each PDB entry 6UNS, 6UNQ, 6UNR and 6UNP?

We have added the title to each PDB entry.

- Supplementary Table 2 lists a number of simulations that we not referenced (as far as I could tell) in the manuscript. In particular, the simulations with the motivation of assessing “stability” of a dimer. Is the text referencing this table inadvertently missing from the main manuscript?

The simulations of the dimers are now referenced in the captions of Supplementary Figs. 2, 10, and 12, which analyze the stability of the ALK2-BMP2 dimers.

Reviewers' Comments:

Reviewer #1:

Remarks to the Author:

The authors have addressed all of my (minor) suggestions. The paper should be published without delay.

Reviewer #3:

Remarks to the Author:

The authors have made the clarifications requested and included supplementary figures that better establish their methods and interpretation of the results. I believe the article to be suitable for publication.